# Dynamic stability of Sgt2 enables selective and privileged client handover in a chaperone triad

Hyunju Cho [1,4], Yumeng Liu [1,5], SangYoon Chung[2], Sowmya Chandrasekar[1], Shimon Weiss[2,3] & Shu-ou Shan [1] ✉

Membrane protein biogenesis poses acute challenges to protein homeostasis, and how they are selectively escorted to the target membrane is not well understood. Here we address this question in the guided-entry-of-tail-anchored protein (GET) pathway, in which tail-anchored membrane proteins (TAs) are relayed through an Hsp70-Sgt2-Get3 chaperone triad for targeting to the endoplasmic reticulum. We show that the Hsp70 ATPase cycle and TA substrate drive dimeric Sgt2 from a wide-open conformation to a closed state, in which TAs are protected by both substrate binding domains of Sgt2. Get3 is privileged to receive TA from closed Sgt2, whereas off-pathway chaperones remove TAs from open Sgt2. Sgt2 closing is less favorable with suboptimal GET substrates, which are rejected during or after the Hsp70-to-Sgt2 handover. Our results demonstrate how fine-tuned conformational dynamics in Sgt2 enable hydrophobic TAs to be effectively funneled onto their dedicated targeting factor while also providing a mechanism for substrate selection.

The correct folding, localization, and quality control of the proteome is a prerequisite for the proper functioning of all cells[1,2]. Among cellular proteins, the biogenesis of integral membrane proteins (MPs) is energetically costly, kinetically demanding, and poses an acute challenge to cellular protein homeostasis. Before arrival at the target membrane, the improper exposure of hydrophobic transmembrane domains (TMDs) on MPs could lead to their rapid and irreversible aggregation in the cytosol. This disrupts the targeting process and generates toxic species that could induce proteostatic stress[3,4]. In addition, MPs delivered to different organelles or by different pathways share substantial overlap in their targeting signals, posing challenging questions about how the specificity of their localization is achieved[5–7]. Therefore, post-translational MP targeting relies critically on molecular chaperones, which must effectively protect newly

synthesized MPs from misfolding and aggregation, and escort them to the target membrane, while also maintaining the selectivity of the targeting pathway.

Tail-anchored membrane proteins (TAs), which contain a single TMD helix near the C-terminus, provide a paradigm for the challenges associated with post-translational MP targeting. TAs comprise 3–5% of the eukaryotic membrane proteome and mediate diverse cellular processes, including protein translocation, vesicular transport, apoptosis, and protein quality control[7,8]. Because their C-terminal TMDs are obscured by the ribosome during translation, TAs are delivered solely by post-translational mechanisms[9]. Multiple pathways for TA targeting the endoplasmic reticulum (ER) and mitochondria have been described, among which the guided-entry-of-tail-anchored protein (GET) pathway is the best studied[3,7,8,10–17]. Biochemical, structural, and yeast

[1]Division of Chemistry and Chemical Engineering, California Institute of Technology, Pasadena, CA 91125, USA. [2]Department of Chemistry and Biochemistry, University of California Los Angeles, Los Angeles, CA 90095, USA. [3]Department of Physics, Institute of Nanotechnology and Advanced Materials, Bar-Ilan University, Ramat-Gan 52900, Israel. [4]Present address: Center for Biomolecular and Cellular Structure, Institute for Basic Science, Daejeon 34126, Republic of Korea. [5]Present address: Biochemistry and Molecular Biotechnology Department, University of Massachusetts Chan Medical School, Worcester, MA 01655, USA. ✉e-mail: sshan@caltech.edu

genetics studies showed that the major cytosolic Hsp70 (Ssa1 in yeast), with the help of Hsp40 (Ydj1 or Sis1 in yeast), effectively captures newly synthesized GET substrates and is important for maintaining them in a soluble, targeting-competent conformation[17–19]. Hsp70 further initiates a sequential series of energetically downhill TA transfers. The first transfer, from Ssa1 to the cochaperone Sgt2, occurs when the two chaperones assemble a complex[18,19]. The second TA transfer, from Sgt2 (or mammalian SGTA) to the ATPase Get3 (or mammalian TRC40), occurs when they are brought together by the Get4/5 scaffold complex[13,20–22]. TA loading activates ATP hydrolysis on Get3 and commits it to deliver TAs to the Get1/2 receptor/insertase complex at the ER membrane[23–27]. Some recent studies suggested an alternative mechanism in which Get4/5-dependent Sgt2 recruitment to the ribosome can allow direct TA capture[28,29]; however, the contribution of this mechanism to TA targeting in vivo is unclear.

The GET pathway provides a salient example to demonstrate that even a compositionally simple MP, such as a TA, is targeted through an elaborate chaperone network[17]. Among the chaperones in this pathway, Sgt2 is the least understood. Sgt2 is an obligate dimer composed of three domains. The N-terminal domain (NTD) mediates its homodimerization (Fig. 1a) and provides an interaction platform for Get4/5, which bridges Sgt2 and Get3 to facilitate TA transfer between them[13,30–34]. The tetratricopeptide repeat (TPR) domain of Sgt2 provides a binding site for the EEVD motif of multiple chaperones, including Hsp70, Hsp90, and Hsp104[13,31,35]. The C-terminal glutamine- and methionine-rich domain comprises the substrate binding domain (SBD) of Sgt2 (Fig. 1a) and shares homology with the Hsp90 cochaperone Sti1/HOP[36,37]. Computational modeling and pulldown experiments suggested that each Sgt2 SBD forms a helical hand presenting a hydrophobic groove that accommodates ~11 amino acids, or roughly half of a TA-TMD[36], raising questions about whether both SBDs in the Sgt2 dimer are required for TA binding. However, previous SAXS (small-angle X-ray scattering) and NMR analyses suggested that the Sgt2 dimer adopts a wide-open and elongated conformation in which the two SBDs are predicted to be 130 Å apart[31,36]. On the other hand, EPR and NMR studies suggested a tendency of the SBDs to dimerize within SGTA[38,39]. The molecular basis of TA recognition by the Sgt2 dimer remains unclear.

Previous studies also yielded contradictory views on the stability of Sgt2's substrate interaction. Using calmodulin (CaM) as a competitor to challenge a preformed SGTA-TA complex, Shao et al. suggested that TA dissociates from SGTA rapidly, with a halftime ($t_{1/2}$) of ~12 sec[40]. In comparison, the transfer of TA from SGTA to TRC40 was ~2-fold faster; the transfer of TA from SGTA to BAG6, which ubiquitylates TA for potential quality control, was approximately twofold slower[40]. These values suggest that SGTA-bound TAs are only modestly favored for GET-dependent targeting over constitutive dissociation and degradation[40]. However, another study measured a TA dissociation rate constant ($k_{off}$) of 0.01–0.02 min$^{-1}$ for the Sgt2-TA complex[18]; this value is more consistent with the observation that the Sgt2-to-Get3 TA transfer is impervious to challenge by CaM[40,41] and suggest that Sgt2-bound TAs are strongly favored for GET-dependent targeting. How Sgt2-bound TAs partition between on- and off-pathway interactions remains an outstanding question.

While the GET pathway targets TAs with hydrophobic TMDs to the endomembrane system, TAs with less hydrophobic TMDs can be delivered by other chaperones, such as CaM, to the ER for insertion by the EMC complex[15]. In addition, Hsp40 and Hsp70 play key roles in delivering multiple MPs, including TAs, to Tom70 on the mitochondrial surface[14,16,42]. How diverse TAs are sorted between different targeting pathways is not well understood. Previous pulldown experiments suggested that Sgt2 may act as an early selection filter in the GET pathway, by favoring TAs with more hydrophobic TMDs[43]. Recent analyses suggested that the Sgt2 SBD recognizes the hydrophobic face of three helical turns in a TMD[36,37]. How this recognition generates specificity remains unclear.

In this work, we address these questions by characterizing the substrate interaction, kinetics, and conformational dynamics of Sgt2. We show that the Sgt2-TA complex is dynamically stable, from which TAs dissociate slowly but can be facilely handed off to other chaperones. This property can be explained by the dynamic rearrangement of Sgt2 between open and closed conformations, and the formation of closed Sgt2 commits TA to the GET pathway. The conformational dynamics of Sgt2 are fine-tuned by the biophysical property of the TA-TMD, allowing Sgt2 to selectively deliver hydrophobic TAs to Get3 while rejecting suboptimal substrates. Our results demonstrate how hydrophobic TAs are channeled through the Hsp70-Sgt2-Get3 chaperone triad and suggest that the Hsp70-Sgt2 pair acts as an early decision point during the biogenesis of diverse TAs.

## Results

### The Sgt2•TA complex displays dynamic stability
Previous studies reported vastly different kinetic stabilities of the Sgt2-TA complex (or the mammalian SGTA-TA complex), with reported $t_{1/2}$ values for TA dissociation ranging from 12 sec to ~40 min[18,40,41]. We first asked if this discrepancy arose from the different chaperones (cpSRP43 vs. CaM) used to chase the Sgt2-TA (or SGTA-TA) complex in previous studies. We used an established TA dissociation assay based on Förster resonance energy transfer (FRET) between a donor dye, coumarin (CM), labeled in the TMD of a model TA substrate Bos1 (TA$^{CM}$)[43] and an acceptor dye, BODIPY-FL (BFL), labeled at the Sgt2 C-terminus (Fig. 1a)[18]. The Sgt2$^{BFL}$-TA$^{CM}$ complex was reconstituted via transfer from Ssa1[18]. After the transfer, Ssa1 and TA co-purified with His$_6$-tagged Sgt2 under physiological salt conditions, whereas a high salt wash (HSW) removed the Ssa1 bound to Sgt2 (Supplementary Fig. 1a, b)[18]. The addition of excess chase chaperone (CaM or cpSRP43) to the purified Sgt2$^{BFL}$-TA$^{CM}$ complex induced the release of TA$^{CM}$, which was monitored in real-time by the recovery of donor fluorescence (Fig. 1a–c). The apparent rate of TA release from Sgt2 differed drastically depending on the chase. 30 μM CaM induced Bos1 release from Sgt2 with an apparent rate constant of 0.91 min$^{-1}$ (Fig. 1b, blue; Supplementary Data 1). This is ~60-fold faster than that induced by 21 μM cpSRP43 (0.017–0.019 min$^{-1}$) and ~2.5-fold slower than Get4/5-mediated TA transfer to Get3 (2.3 min$^{-1}$; Fig. 1b, blue vs. black and orange). Unlabeled Sgt2 also induced rapid TA release at rates much higher than that observed with cpSRP43 (Fig. 1c, blue).

To understand this behavior, we carried out kinetic simulations for different mechanisms by which the chase chaperones induce TA release from Sgt2. In the simplest model, the chase acts solely as a trap to sequester the free TAs that have dissociated from Sgt2 (Fig. 1d). This model predicts that observed TA release is rate-limited by spontaneous Sgt2-TA dissociation and independent of chase concentration (Fig. 1d). Alternatively, the chase chaperone could 'invade' the Sgt2-TA complex to directly remove TA from Sgt2; this model predicts that TA release will be accelerated by increasing chase concentrations (Fig. 1e). These models can therefore be distinguished by determining the kinetics of TA release at different chase concentrations. The results showed that the same rate constant of TA release was observed at different cpSRP43 concentrations (Fig. 1g, black), indicating that cpSRP43 acted solely as a trap for dissociated TAs and allowed the determination of the intrinsic dissociation rate of the Sgt2-TA complex (Fig. 1d). In contrast, TA release from Sgt2 was strongly accelerated by increasing CaM concentrations (Fig. 1f, g, blue), consistent with the invasion model (Fig. 1e). Extrapolation of the CaM concentration dependence to the Y-axis intercept gave an intrinsic $k_{off}$ value of 0.018 min$^{-1}$, identical to that measured using cpSRP43 as the chase (Fig. 1g).

Together, these results show that the Sgt2-TA complex has high intrinsic kinetic stability but can be invaded by other chaperones, such that the bound TA is readily handed off to different Sgt2 molecules or to other TA-binding chaperones, such as CaM, without releasing free TAs to solution. We term this behavior 'dynamic stability'.

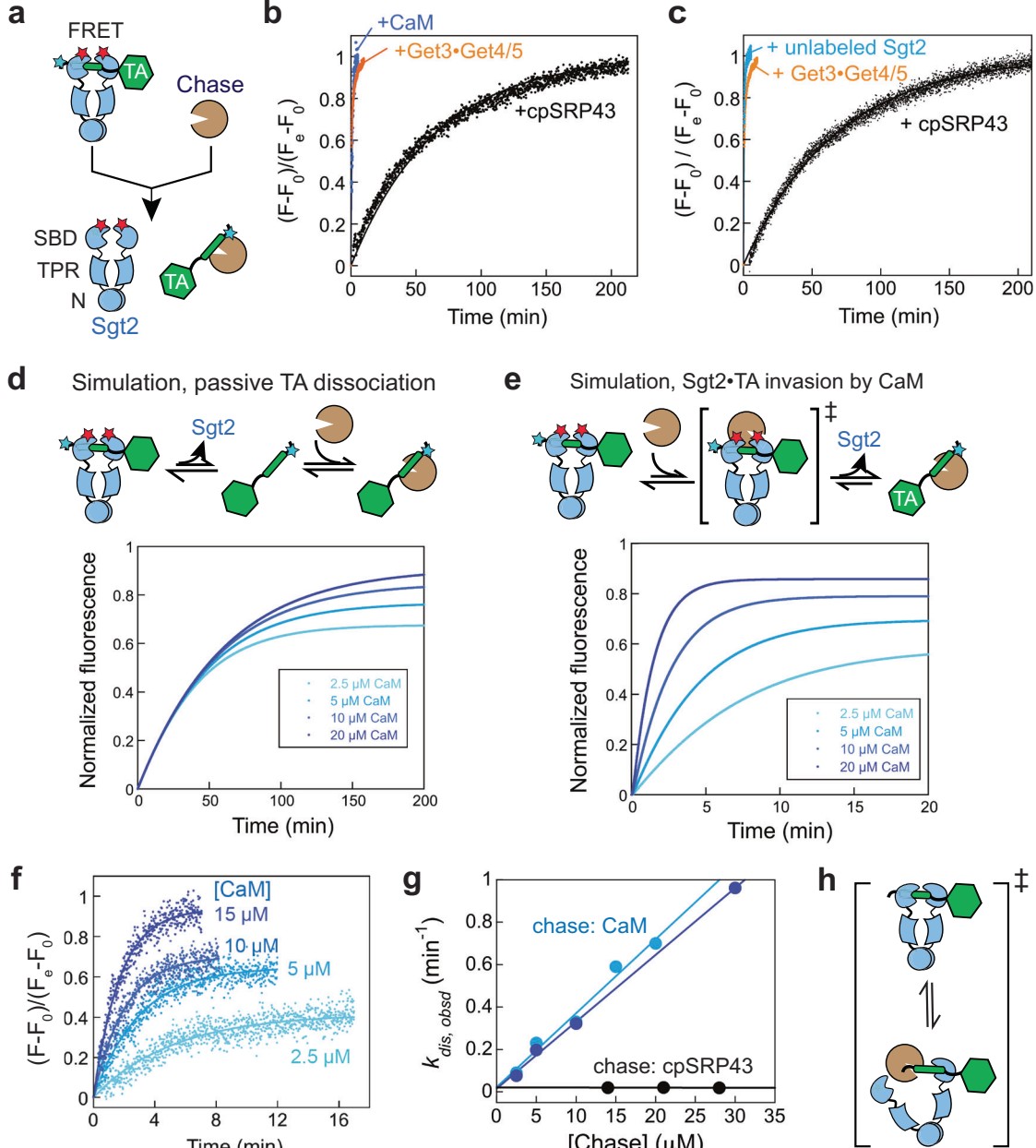

**Fig. 1 | The Sgt2-TA interaction displays dynamic stability. a** Scheme of the FRET assay to measure TA release from Sgt2. Cyan and red stars denote the donor and acceptor dyes on TA and Sgt2, respectively. Addition of excess chase to a purified Sgt2•TA complex initiates TA release and leads to loss of FRET. **b**, **c** Distinct kinetics of TA loss from Sgt2 with different chase chaperones (Sgt2, CaM, or cpSRP43). The kinetics of TA transfer to the Get3•Get4/5 complex (*orange*) is shown for comparison and was measured using a FRET pair between TA and Get3[43]. Fluorescence time traces are normalized such that $F = 0$ at $t = 0$, and F = 1 at the end of the reaction. Two representative data with cpSRP43 as chase (*black*) from four independent replicates are shown and were fit to Eq. 1, which gave rate constants of 0.017 min⁻¹ in (**b**) and 0.018 min⁻¹ in (**c**). The data with CaM, unlabeled Sgt2, and Get3/4/5 were fit to Eq. 2. **d**, **e** Simulated kinetics of TA loss from Sgt2 in models where TA spontaneously dissociates from Sgt2 (**d**) or where an external chaperone removes TA from Sgt2 (**e**). **f** TA loss from Sgt2 is accelerated by increasing CaM concentration. Fluorescence time traces are normalized such that $F = 0$ at $t = 0$, and $F = 1$ at the end of the reaction with 15 μM CaM. The lines are fits of the data to Eq. 2. A representative dataset from duplicate measurements is shown. **g** The observed rate constant of TA release from Sgt2 as a function of chase (CaM or cpSRP43) concentration. Light and dark blue show two independent sets of measurement with CaM. Extrapolation of both data to the Y-axis gives the intrinsic $k_{off}$ of 0.018 min⁻¹, identical to measurements using three different concentrations of cpSRP43. **h** Model to explain the effect of CaM on TA release rates. CaM invades the Sgt2-TA complex by interacting with a partially exposed TA-TMD. Source data are provided in the Source Data file.

## Both SBDs of an Sgt2 dimer are required for high-affinity TA binding

In seeking a molecular explanation for the dynamic stability of the Sgt2-TA interaction, we considered the fact that Sgt2 is an obligate homodimer. Previous studies suggest that each SBD of Sgt2 accommodates three α-helical turns, or 11 amino acids[36], whereas the TMD of an ER-destined TA contains ~20 amino acids. We therefore

hypothesized that complete protection of the TA-TMD requires interaction with both SBDs in Sgt2. Engagement with only one of the SBDs could expose part of the TA-TMD for interaction with other chaperone molecules, potentially leading to TA exchange (Fig. 1h).

This model requires that both SBDs of Sgt2 engage in high-affinity TA binding. To test this, we used a mutant Sgt2ΔN (96–346 aa) in which the N-terminal dimerization domain is deleted

## a

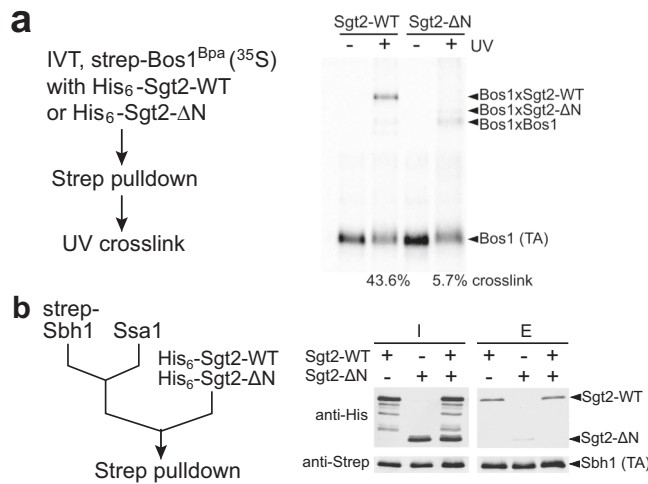

IVT, strep-Bos1$^{Bpa}$ ($^{35}$S)
with His$_6$-Sgt2-WT
or His$_6$-Sgt2-ΔN

↓

Strep pulldown

↓

UV crosslink

## b

strep-Sbh1  Ssa1

His$_6$-Sgt2-WT
His$_6$-Sgt2-ΔN

↓

Strep pulldown

**Fig. 2 | Both SBDs in Sgt2 are required for high-affinity TA binding. a** Bpa-incorporated TA (Bos1) was translated in the S30 extract in the presence of 2 μM His$_6$-tagged wildtype Sgt2 or Sgt2ΔN. TA association with Sgt2 was analyzed by UV crosslinking and autoradiography after pulldown on strep$_3$-tagged TA. **b** Recombinant Strep-tagged Sbh1 (0.75 μM) was preincubated with 6 μM Ssa1 in the presence of 2 mM ATP, followed by the addition of 3 μM Sgt2, Sgt2ΔN, or both. TA was captured using streptactin resin, and the associated Sgt2 or Sgt2ΔN was detected by western blot analysis. Representative data were shown in **a** and **b** with two independent experiments (*n* = 2). Source data are provided as a Source Data file.

(Supplementary Fig. 2a, b). We first compared Sgt2-WT and Sgt2ΔN in a TA capture assay in which the TA substrate Bos1 was generated by in vitro translation (IVT) in the *E. coli* S30 extract devoid of GET pathway components (Fig. 2a)[43,44]. An unnatural amino acid photocrosslinker, p-benzoylphenylalanine (Bpa), replaced Ala228 at the 6th residue in the Bos1-TMD during IVT[18,45,46]. The inclusion of Sgt2 during IVT allowed the formation of a Sgt2-TA complex, which was purified via the strep tag on Bos1. TA association with Sgt2 was monitored by UV-induced crosslinking of Bpa. Over 43% of Bos1$^{Bpa}$ crosslinked to Sgt2-WT, whereas ~6% of Bos1$^{Bpa}$ crosslinked to Sgt2ΔN (Fig. 2a). Thus, dimerization of Sgt2 is required for stable TA interaction with Sgt2.

To independently test this model, we prepared the Sgt2-TA complex via transfer from Ssa1 using another model TA, Sbh1 (Fig. 2b, left panel, and Supplementary Fig. 2c). TA association with Sgt2 was monitored using pulldown on strep-tagged TA and western blot against Sgt2 or Sgt2ΔN. Significantly less Sgt2ΔN co-purified with TA than Sgt2-WT (Fig. 2b, right panel). Furthermore, when an equimolar mixture of Sgt2-WT and Sgt2ΔN was presented to the preformed Ssa1-Sbh1 complex, Sgt2ΔN was out-competed by Sgt2-WT and not detectable after TA pulldown (Fig. 2b, right panel). These results show that the dimerization of Sgt2 is indispensable for efficient and stable TA capture by Sgt2, suggesting that both SBDs in the Sgt2 dimer engage the TA-TMD.

### Conformational dynamics of Sgt2 during successive TA transfers

Previous SAXS and other biophysical analyses revealed an extended structure of free dimeric Sgt2 in which its two SBDs are 130 Å apart[31,39]. However, the observation above suggests that conformational changes in Sgt2 must occur upon TA binding to bring the two SBDs into close proximity (Fig. 1h). The ability of the SBDs to dimerize has been suggested for SGTA in EPR and NMR studies[38]. To test this model directly, we measured intradimer FRET between the SBDs of Sgt2. Dimeric Sgt2 was doubly labeled with donor (ATTO550) and acceptor (ATTO647N) dyes at the C-terminus, such that closed Sgt2 will display

high FRET, whereas a wide-open Sgt2 is expected to display low FRET (Fig. 3a).

Steady-state fluorescence measurements showed that doubly labeled apo-Sgt2 displayed minimal FRET. Upon excitation of the donor dye, the acceptor-to-donor fluorescence intensity ratio was 0.11 (Supplementary Fig. 3a, b). This ratio increased to 0.23 when TA was loaded onto Sgt2 via transfer from Ssa1 (Fig. 3b, 1st transfer, and Supplementary Fig. 3a, b). Upon further addition of Get3 and Get4/5 to induce TA transfer from Sgt2 to Get3 (Fig. 3b, 2nd transfer), this ratio returned to 0.12 (Supplementary Fig. 3a, b). Control experiments showed that nucleotide, TA substrate, chaperones (Ssa1, Ydj1), and additional interaction partners (Get3, Get4/5) did not significantly affect the fluorescence emission spectra of either dye (Supplementary Fig. 3c, d). Thus, the observed relative fluorescence changes of the acceptor and donor dyes can be attributed to conformational rearrangements of Sgt2.

To resolve labeling and conformational heterogeneity and directly observe the conformational dynamics of Sgt2, we carried out diffusion-based single molecule (sm)FRET measurements using Alternating Laser Excitation Spectroscopy with microsecond resolution (μs-ALEX). ALEX allows the optical purification of doubly labeled Sgt2 molecules by removing contributions from singly labeled species[47,48] and facilitates the determination of the conformational distribution of Sgt2 (Fig. 3a). The results showed that the smFRET histogram of apo-Sgt2 is dominated by a low FRET population that peaks at an apparent FRET efficiency (*E\**) of ~0.15, consistent with previous SAXS analysis[31] that indicated a wide-open apo-Sgt2 (Fig. 3c, black). After the first TA transfer from Ssa1 to Sgt2 (Fig. 3b), the smFRET histogram shifted rightwards, with an additional new peak at *E\** ~0.85 (Fig. 3c, blue), indicating that a fraction of Sgt2 molecules adopted a closed conformation in which the two SBDs are brought into proximity. Upon further addition of Get3 and Get4/5 (Fig. 3b, 2nd transfer), the histogram was again dominated by a low FRET population (Fig. 3d, red). Neither Get3 nor Get4/5 alone induced these changes (Fig. 3d, yellow and orange), suggesting that Get4/5-dependent TA transfer from Sgt2 to Get3 is responsible for the re-opening of Sgt2.

To investigate whether Sgt2 dynamically samples different conformations on timescales faster than that of diffusion through the observation volume (~milliseconds (ms)), we implemented Burst variance analysis (BVA), which allows the detection of millisecond or faster dynamics by comparing the empirical standard deviation (SD) of *E\** of sub-bursts within a burst to the expected SD from the shot-noise limit[49]. If the FRET distribution arises solely from static species that exchange on timescales much slower than diffusion, the SD will lie on the static limit curve (Supplementary Fig. 4, dashed lines). In contrast, if multiple conformations interconvert on the milliseconds or faster timescale, the observed SD would be higher than the static limit. Deviations from the static limit were found for Sgt2 at intermediate *E* values under all conditions tested (Supplementary Fig. 4), indicating that the intermediate E values arose from dynamic transitions of Sgt2 between the open and closed conformations during each burst.

To quantitatively characterize these transitions, we performed multi-parameter (mp) photon-by-photon Hidden Markov modeling (H$^2$MM) analysis (Supplementary Fig. 5)[50,51]. H$^2$MM enables HMM analysis for diffusion-based smFRET using a modified maximum likelihood estimator, which applies HMM algorithm directly to photon data and successfully retrieves information about the conformational dynamics of biomolecules ranging from sub-seconds to microseconds[51]. mpH$^2$MM further distinguishes subpopulations that represent conformational states (varying E values with *S* ~ 0.5) versus those that arise from photophysical artifacts, such as acceptor blinking (*E* ~ 0 and *S* ~ 1) and thus allows more reliable extraction of state parameters (mean E values and rate constants)[50]. mpH$^2$MM analysis showed that all the μs-ALEX data for Sgt2 and Sgt2 complexes could be accounted for by a three-state model, with a low FRET (*E*$_{mean}$ ~0.2) and a high FRET state

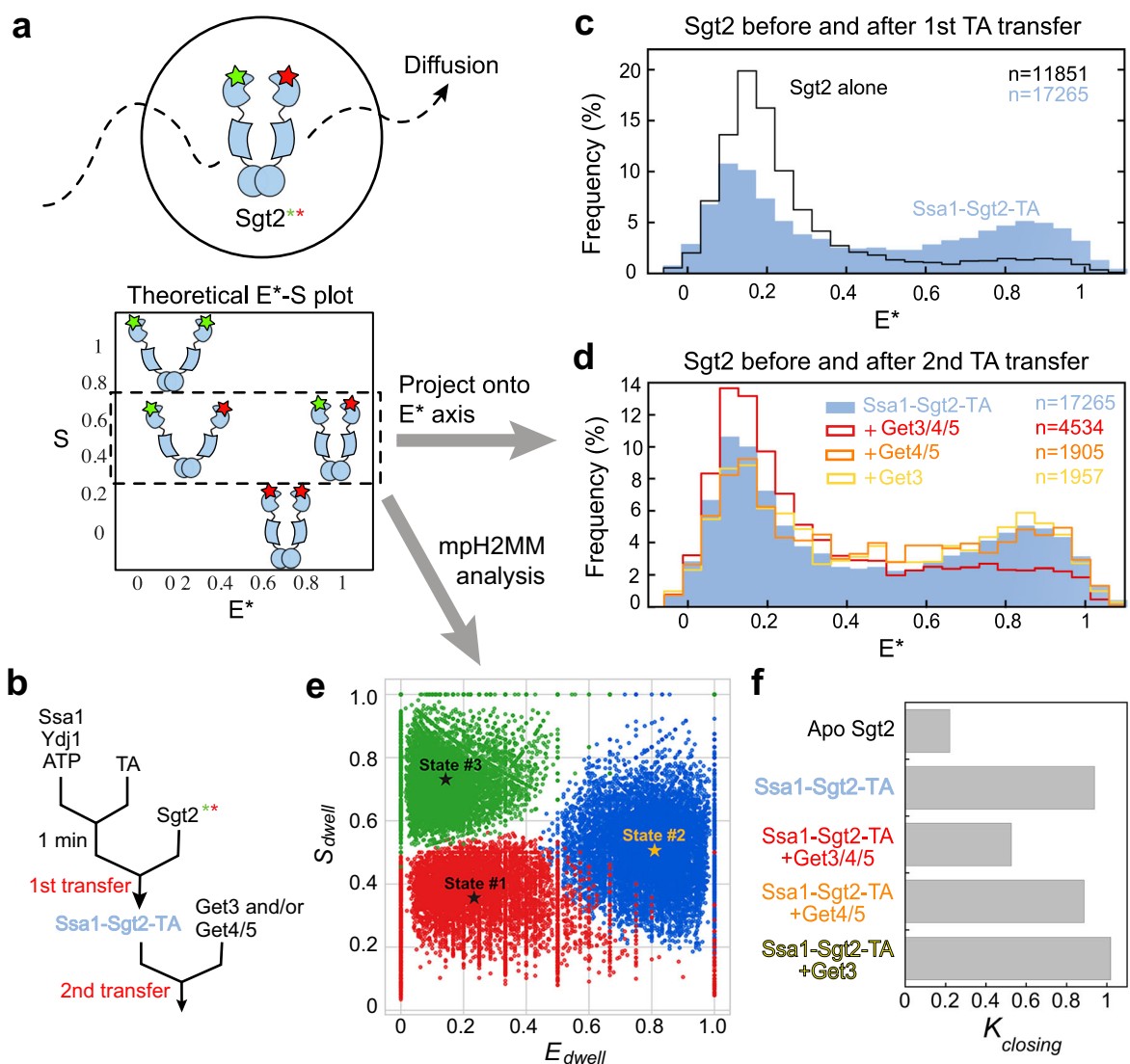

**Fig. 3 | smFRET detects conformational changes of Sgt2 during sequential TA transfers in the GET pathway. a** Scheme of the μsALEX-smFRET measurements to determine the conformational distribution of Sgt2 in solution. Calculation of donor stoichiometry (*S*) allows optical purification of Sgt2 molecules labeled with both donor and acceptor dyes (*S*-0.5). Apparent FRET efficiency (*E*\*) reflects changes in the distance between the dye pair for Sgt2 in distinct conformations. **b** Sample preparation workflow for the experiments in **c** and **d**. 0.1 μM TA was loaded onto 3 μM Ssa1 in the presence of 3 μM Ydj1 and 2 mM ATP, followed by the addition of 100–200 pM doubly labeled Sgt2 to allow the Ssa1-to-Sgt2 TA transfer (1st transfer). Where indicated, 0.5 μM Get3 with or without 0.5 μM Get4/5 was added to allow TA

transfer from Sgt2 to Get3 (2nd transfer). **c, d** smFRET histogram of apo-Sgt2 and its complex after the first TA transfer from Ssa1/Ydj1 to Sgt2 (**c**) and of Sgt2 before and after the second TA transfer (**d**). '*n*' indicates the number of data points used to construct the histograms. **e** Results of mpH²MM analysis of the sample after the 1st TA transfer. States 1 and 2 are open and closed conformations of Sgt2 with low and high *E*\* values, respectively, and state 3 arises from dye photophysics, such as acceptor blinking. Rate and equilibrium constants from the H2MM analyses are summarized in Table S1 and Supplementary Data 1. **f** Summary of the equilibrium constants for Sgt2 closing (*K*$_{closing}$) based on the mpH²MM analyses.

($E_{mean}$ ~0.8) that interconverts on the millisecond timescale, and a third sub-population that arose from dye photophysical events such as acceptor blinking (Fig. 3e and Supplementary Data 1). The interconversion rate constants further allowed us to calculate the equilibrium for the open-to-closed rearrangement (*K*$_{closing}$; Table S1), which would be challenging to obtain by directly fitting the smFRET histogram. Based on the mpH²MM analysis, *K*$_{closing}$ is unfavorable in apo-Sgt2 (0.221), rose to 0.937 after the Ssa1-to-Sgt2 TA transfer, and reduced to 0.525 after Get4/5-dependent TA transfer from Sgt2 to Get3 (Fig. 3e, f and Table S1). In addition, *K*$_{closing}$ after Get4/5-dependent TA transfer to Get3 is similar to that of Sgt2 in the presence of Ssa1, Ydj1, and ATP (0.525 vs. 0.473; Table S1) and much higher than that for apo-Sgt2, suggesting that Ssa1 remains bound to Sgt2 after the second TA transfer.

Together, these results provide direct evidence that Sgt2 dynamically samples open and closed conformations. A substantial population of Sgt2 molecules assumes the closed conformation after Ssa1/Ydj1-mediated TA loading and resumes the open conformation upon further transfer of the TA to Get3.

**The TA substrate and upstream chaperones drive Sgt2 closing**
Since the first TA transfer generates an Ssa1-Sgt2-TA complex (Supplementary Fig. 1a, b), the observed Sgt2 closing could arise from Ssa1 binding, TA loading, or both. To dissect the contribution of TA, we purified the Sgt2-TA complex (Supplementary Fig. 1a, b). μs-ALEX measurements showed that the TA substrate alone increased *K*$_{closing}$ from 0.221 to 0.597 (Fig. 4a, d, dark green; Table S1). This indicates that the TA substrate can by itself drive

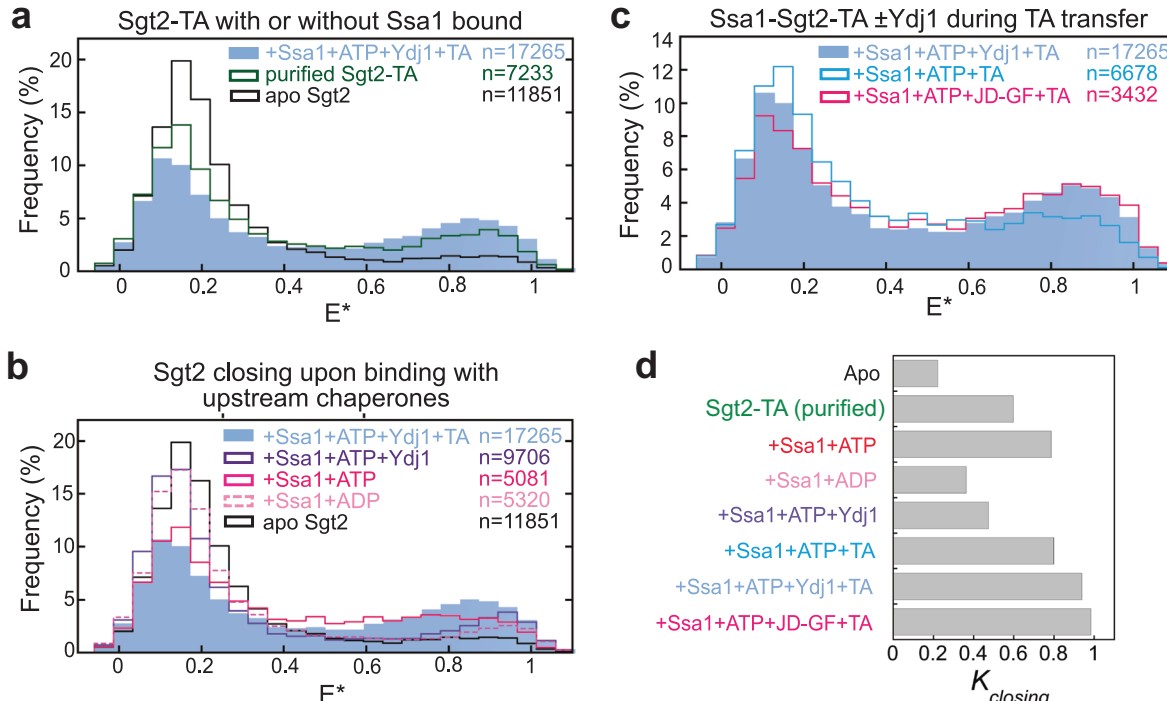

**Fig. 4 | Both the TA substrate and upstream chaperones are required to maximize Sgt2 closing. a** smFRET histogram of purified Sgt2-TA complex. The data for apo-Sgt2 and the Ssa1-Sgt2-TA complex are from Fig. 3c and shown for comparison. **b** smFRET histograms of Sgt2 in the presence of 3 μM Ssa1, 2 mM ATP or ADP, with and without 3 μM Ydj1. The data for apo-Sgt2 and the Ssa1-Sgt2-TA complex are shown for comparison. **c** smFRET histogram of the Ssa1•Sgt2•TA complex formed in the presence and absence of Ydj1 variants. **d** Summary of the effects of TA and upstream chaperones on the equilibrium of Sgt2 closing obtained from mpH²MM analyses (Table S1 and Supplementary Data 1).

substantial Sgt2 closing, and reciprocally, *closed* Sgt2 binds TA with higher affinity than *open* Sgt2.

Remarkably, ATP-bound Ssa1 also substantially enhanced Sgt2 closing ($K_{closing}$ = 0.785) and accelerated the open-to-closed transition, whereas ADP-bound Ssa1 induced much smaller changes (Fig. 4b, d, red vs. light pink; Table S1). The additional presence of Ydj1, which activates ATP hydrolysis on Ssa1, exerted an effect that is intermediate between Ssa1[ATP] and Ssa1[ADP] (Fig. 4b, d, purple; Table S1). These results strongly suggest that ATP-bound Ssa1 induces Sgt2 to sample the closed conformation, potentially priming it for TA capture.

We previously showed that activation of Ssa1 ATPase activity by Ydj1 or a fragment containing the Ydj1 J-domain (JD-GF) enhances the crosslinking efficiency of TA[Bpa] to Sgt2[19]. However, pulldown experiments showed that the same amount of TA co-purified with Sgt2 regardless of whether Ydj1 was present during the Ssa1-to-Sgt2 TA transfer (Supplementary Fig. 1c, d), excluding the possibility that Ydj1 increased the amount of TA loading on Sgt2. To test whether the effect of Ydj1 was due to changes in Sgt2 conformation, we carried out the Ssa1-to-Sgt2 TA transfer without Ydj1 (Fig. 4c). The resulting Ssa1-Sgt2-TA complex is compositionally identical to that generated with Ydj1 present, as Ydj1 does not stably associate with the Ssa1-Sgt2-TA complex ([19] and Supplementary Fig. 1b). Nevertheless, the complex generated without Ydj1 showed reduced Sgt2 closing ($K_{closing}$ = 0.798 vs 0.937; Fig. 4c, d, cyan vs light blue, and Table S1). The JD-GF fragment of Ydj1, which contains only the J-domain that activates ATP hydrolysis of Ssa1[19], was sufficient to replace Ydj1 and induce maximal Sgt2 closing ($K_{closing}$ = 0.983; Fig. 4c, d, hot pink; Table S1). Thus, Ydj1-induced ATPase activation of Ssa1 maximizes Sgt2 closing in the Ssa1-Sgt2-TA complex. Combined with the enhanced TA crosslinking efficiency to Sgt2 in the presence of Ydj1[19], these results suggest that the ATPase cycle of Ssa1 optimizes the conformation of Sgt2 for TA interactions.

## A closed Sgt2 prevents TA loss and selectively delivers the TA to Get3

To understand the functional implications of Sgt2 closing, we sought to lock Sgt2 in the closed state by introducing an intradimer crosslink that brings its two SBDs into proximity. To this end, we engineered single cysteines in the Sgt2 SBD and the TPR-SBD linker (Supplementary Fig. 6a). Sgt2 single cysteine mutants are functional in capturing TA from Ssa1, as assessed by UV-induced crosslink of TA[Bpa] to Sgt2 (Supplementary Fig. 6b, c). We then introduced an intradimer crosslink in each Sgt2 variant using the thiol-specific homo-bifunctional crosslinker bismaleimidohexane (BMH). BMH crosslinking efficiency was highest (46%) for Sgt2-C258, in which the engineered cysteine is two residues N-terminal to the SBD (Supplementary Fig. 6d) and was therefore used in subsequent experiments. We further verified that the BMH-mediated crosslink occurred within, but not between, Sgt2 dimers (Supplementary Fig. 7a).

To resolve the interaction of TA with different chaperones and with Sgt2 in different conformations, we used TA[Bpa] crosslinking followed by SDS-PAGE (Supplementary Fig. 7b). Sgt2-TA[Bpa] complexes were generated by IVT of TA[Bpa] in S30 lysate in the presence of wildtype Sgt2 or Sgt2-C258, purified via the His$_6$-tag on Sgt2, and crosslinked with BMH (Fig. 5a). This resulted in ~50% of Sgt2-C258 crosslinked by BMH, which was denoted as (Sgt2-C258)$_{XL2}$ (crosslinked Sgt2 dimer; Supplementary Fig. 7c). Upon subsequent UV irradiation, substantially more TA[Bpa] crosslinked to (Sgt2-C258)$_{XL2}$ than to Sgt2-C258 (Fig. 5c, bands at ~120 kDa vs ~65 kDa, *t* = 0), providing additional support that TA preferentially interacts with closed Sgt2.

Using this workflow (Fig. 5a), we tested whether and how the TA bound to BMH crosslinked (Sgt2-C258)$_{XL2}$ (representing closed Sgt2 dimer) and uncrosslinked Sgt2-C258 (representing Sgt2 sampling open and closed conformations) is susceptible to loss to external chaperones. In agreement with FRET measurements (Fig. 1) and previous

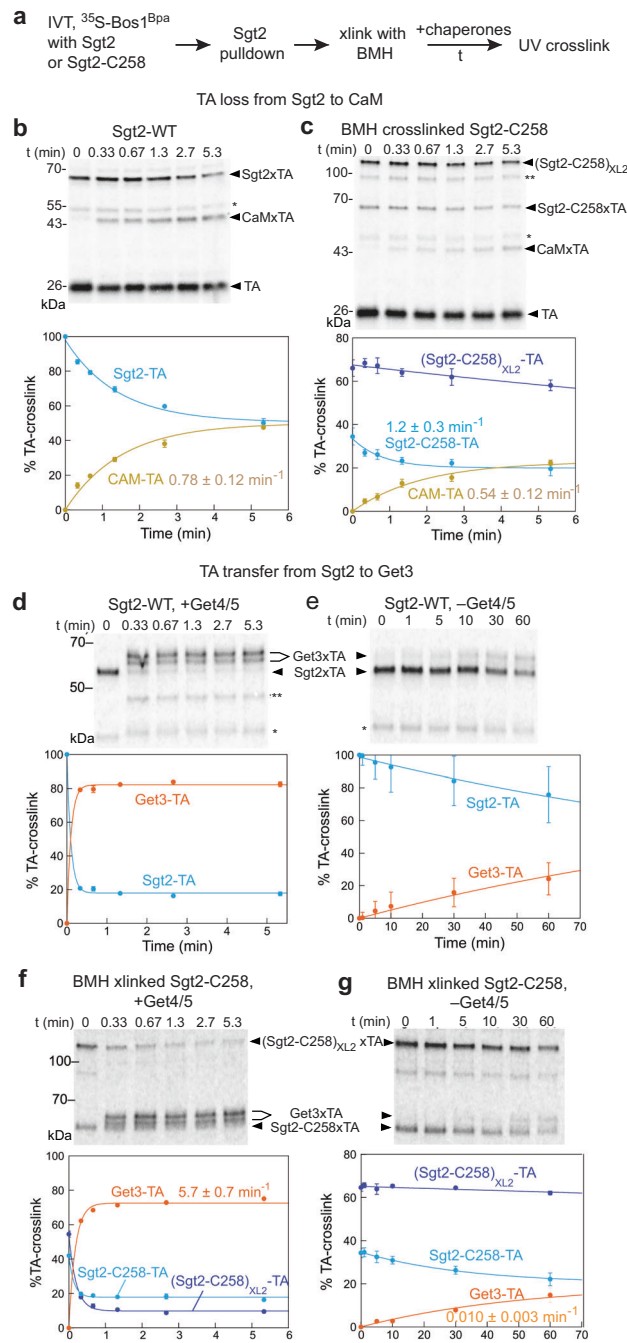

**Fig. 5 | Crosslinked Sgt2 dimer protects TA loss but can transfer TA to Get3.**
**a** Summary of experimental workflow, as described in the text and Methods.
**b**, **c** The kinetics of TA loss to CaM was measured for wildtype Sgt2-TA (**b**) and BMH-crosslinked Sgt2-C258-TA (**c**). **d**–**g** Kinetics of TA transfer to Get3 was measured for Sgt2-TA (**d**, **e**) and BMH-crosslinked Sgt2-C258-TA (**f**, **g**) in the absence (**e**, **g**) and presence (**d**, **f**) of Get4/5. **b**–**g** Upper panels show a representative SDS-PAGE autoradiograph, and lower panels show the quantification of triplicate data. * and ** denote minor crosslinked species in the presence of Sgt2 or Get3 that were not interpreted. Data are represented as mean ± SEM (*n* = 3). Lines are fits of the data to single exponential functions. The obtained rate constants are indicated where appropriate. Source data are provided in the Source Data file.

work[40,41], wildtype Sgt2-TA$^{Bpa}$ lost the TA to CaM quickly, with an apparent rate constant of 0.78 min$^{-1}$ in the presence of 20 μM CaM (Fig. 5b). In contrast, the association of TA with (Sgt2-C258)$_{XL2}$ remained largely intact when challenged by CaM (Fig. 5c). Although a small amount of TA-CaM complex accumulated over time,

quantification of the data suggested that the majority of them, especially those that formed early (<1 min), arose from the TA bound to Sgt2-C258 (Fig. 5c). These results strongly suggest that the TA bound to (Sgt2-C258)$_{XL2}$ is protected from CaM and that kinetically facile TA loss from Sgt2 primarily occurs from open Sgt2.

We next asked if Sgt2 opening is required for TA transfer from Sgt2 to Get3. To this end, we presented Get3 with and without Get4/5 to uncrosslinked and BMH-crosslinked Sgt2-C258-TA$^{Bpa}$ complexes. With wildtype Sgt2-TA$^{Bpa}$, TA transfer to Get3 was complete in 20 sec and depended strongly on Get4/5 (Fig. 5d, e). Unexpectedly, TA transfer from (Sgt2-C258)$_{XL2}$ to Get3 was also rapid in the presence of Get4/5: the TA crosslink to (Sgt2-C258)$_{XL2}$ disappeared at a rate constant of 3.5 min$^{-1}$, with the concurrent appearance of TA crosslink to Get3 (Fig. 5f). This transfer was abolished in the absence of Get4/5, and the majority of Get4/5-independent transfer to Get3 arose slowly from the TA bound to uncrosslinked Sgt2-C258 (Fig. 5g). These results show that the TA bound to closed Sgt2 is protected from off-pathway chaperones and other mechanisms of nonspecific loss from Sgt2, and only Get3 is uniquely privileged to capture the TA from closed Sgt2 in the transfer complex assembled by Get4/5.

### Facilitated TA transfer from Ssa1 to Sgt2

The ability of Ssa1/Ydj1 to induce Sgt2 closing raised questions about whether the Ssa1-to-Sgt2 TA transfer is also privileged, similar to the Sgt2-to-Get3 transfer. To address this question, we compared the kinetics of this transfer with spontaneous TA dissociation from Ssa1. We developed a FRET-based assay to detect the interaction of TA with Ssa1 using BFL as the donor dye labeled near the TA-TMD and tetramethylrhodamine (TMR) as the acceptor dye labeled at an engineered single cysteine D430C in the Ssa1 SBD (Fig. 6a). Thio-substitution and fluorescence labeling at D430C does not affect the activity of Ssa1[18,52]. TMR-labeled Ssa1, but not unlabeled Ssa1, reduced the fluorescence intensity of BFL-labeled Bos1 ~20%, and this change was reversed by a five-fold excess of unlabeled Ssa1 (Fig. 6b), demonstrating FRET between the dye pair.

Using this assay, we measured the kinetics of TA dissociation from Ssa1 by chasing a preformed Ssa1$^{TMR}$•Bos1$^{BFL}$ complex with excess cpSRP43. The time course of TA release from Ssa1 was biphasic, with ~50% of the fluorescence change occurring at a rate constant of 0.63–0.80 min$^{-1}$ and the other ~50% at ~0.03 min$^{-1}$. Similar TA release kinetics were observed with and without Ydj1 present or with different concentrations of cpSRP43 (Fig. 6c, d), indicating that the intrinsic TA dissociation rate from Ssa1 was measured and is unaffected by Ydj1. The biphasic kinetics could arise from two populations of Ssa1-TA complex with different kinetic stabilities or a two-step TA dissociation mechanism, and these possibilities remain to be resolved.

Regardless of the precise interpretation, the rates of both phases during Ssa1-TA dissociation were much slower than the transfer of TA from Ssa1 to Sgt2, which was measured using the Bpa crosslinking assay on a quench-flow device (Fig. 6e). Although the time resolution of this experiment was likely limited by quenching of the transfer reaction (flash freezing in dry ice-ethanol bath, which takes ~1 s), the transfer was complete within 1 sec (Fig. 6e). This is ~100-fold faster than the fast phase and ~2000-fold faster than the slow phase during TA dissociation from Ssa1 (Fig. 6e vs d), indicating that the Ssa1-to-Sgt2 TA transfer does not occur via the dissociation of TA from Ssa1 followed by rebinding to Sgt2.

To independently test this model, we tested whether the Ssa1-to-Sgt2 TA transfer is protected from external chaperones by comparing transfer reactions in the absence and presence of excess CaM. The presence of CaM induced negligible changes in the amount of TAs loaded on Sgt2 (Fig. 6f, g). As a negative control, we also tested a mutant Sgt2(R171A, R175A) in which the interaction of the TPR domain with Ssa1 is disrupted (Sgt2-TPRmt[13,18]). TA transfer from Ssa1 to Sgt2-TPRmt was significantly lower than that to wildtype Sgt2 and was

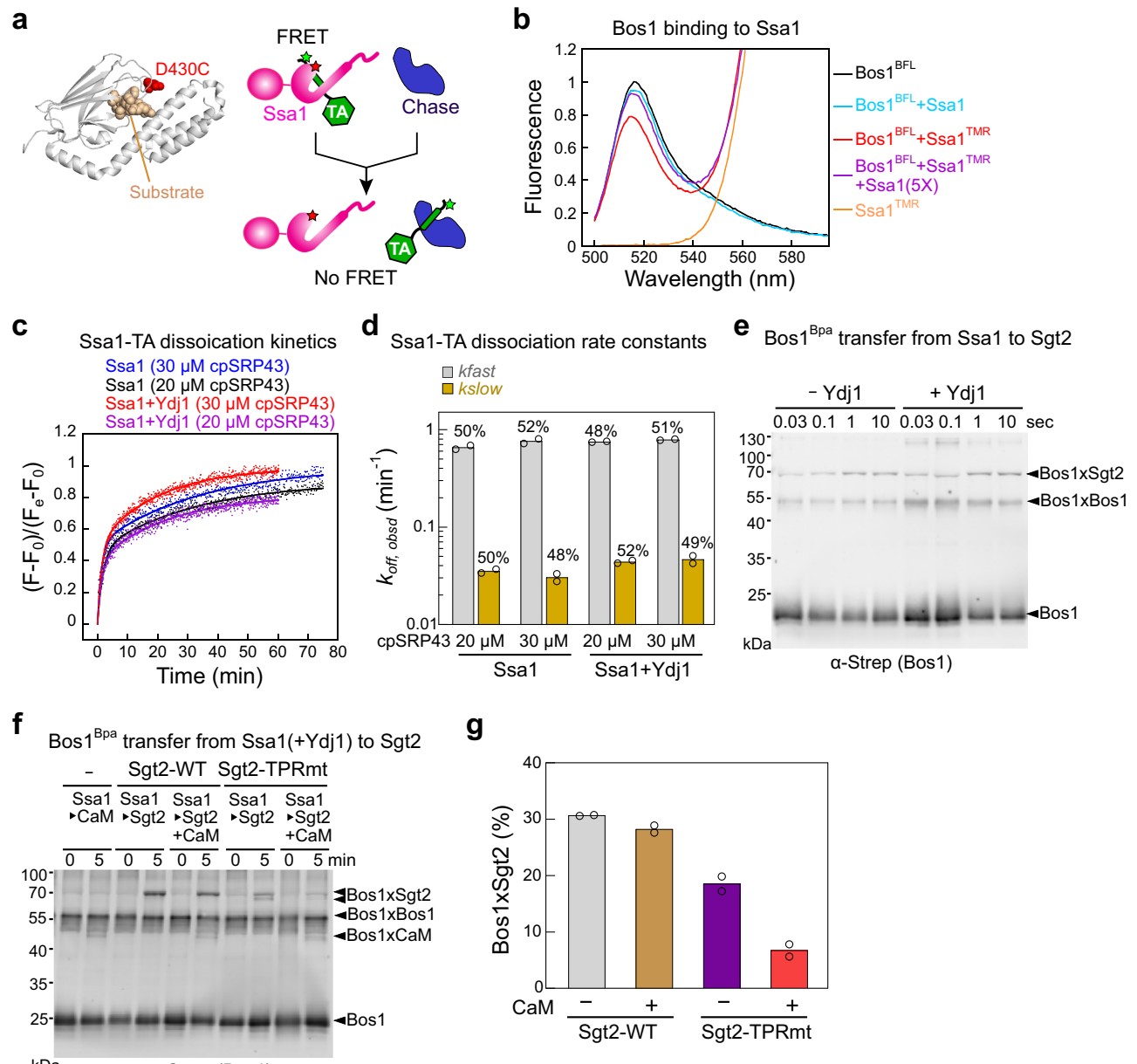

**Fig. 6 | TA transfer from Ssa1 to Sgt2 is privileged. a** Scheme of the FRET assay to measure TA dissociation kinetics from Ssa1. The position of fluorescence labeling is indicated by the structure of the DnaK SBD with a bound peptide (PDB 4EZO). **b** Fluorescence emission spectra showing FRET between BFL-labeled TA and TMR-labeled Ssa1(D430C). 20 nM Bos1[BFL] was incubated with 500 nM Ssa1[TMR] for 5 min, and the donor fluorescence emission spectra were recorded using an excitation wavelength of 485 nm. **c** Representative fluorescence time traces for TA dissociation from Ssa1 with or without Ydj1. 20 nM Bos1[BFL] and 500 nM Ssa1[TMR] were pre-incubated with or without 500 nM Ydj1 for 5 min, followed by the addition of cpSRP43 at indicated concentrations. Data are normalized such that $F = 0$ at $t = 0$, and $F = 1$ at the end of the reaction. Lines are fits of the data to Eq. 2 in Methods. **d** Summary of the TA dissociation rate constants from (C). The mean of two technical replicates is shown. **e** Time courses of TA transfer from Ssa1 to Sgt2 was measured with and without Ydj1 using Bpa crosslinking. **f, g** TA transfer from Ssa1 to Sgt2 was measured with and without CaM for wildtype Sgt2 and Sgt2-TPRmt. Representative western blots are shown in **f**, and the efficiency of the Sgt2-TA crosslink after the transfer is summarized in **g**. Data are shown as mean with $n = 2$. Source data are provided in the Source Data file.

sensitive to CaM (Fig. 6f, g; also[18]), indicating that protected TA transfer requires Ssa1 and Sgt2 to form a complex. Together, these results show that TAs are channeled from Ssa1 to Sgt2 in a highly protected manner.

**The stability and conformation of Sgt2-TA are regulated by biophysical properties of the TA-TMD**

Given the susceptibility of open Sgt2 to external chaperones, we asked if the open conformation played a role in TA selection. Previous pull-down studies in S30 lysate showed that TAs containing TMDs with lower hydrophobicity and helical propensity co-purified less efficiently with Sgt2[43]. To dissect the molecular step(s) responsible for this discrimination, we tested the substrate dependence of the three steps in the generation and maintenance of the Sgt2-TA complex: TA capture by Ssa1, TA transfer from Ssa1 to Sgt2, and TA dissociation from Sgt2. We used an established series of TA variants in which an increasing number of hydrophobic residues in the Bos1-TMD was replaced by Ala and Gly, which reduce the hydrophobicity and helical propensity of the TMD (Supplementary Fig. 8a, 4AG, 5AG, and 6AG)[43]. 4AG and 5AG displayed reduced efficiencies of GET-dependent insertion into the ER,

whereas the ER insertion of 6AG was GET-independent[43]. As an additional negative control, the Bos1-TMD was replaced with that of Fis1, a mitochondrial TA (Supplementary Fig. 8a).

Using a turbidity-based assay that monitors TA aggregation[18], we first showed that Ssa1 efficiently captured and solubilized all the TA variants tested (Supplementary Fig. 8b, c). The solubilization constants of Ssa1 for different TAs varied by less than threefold and did not correlate with their GET-dependence for insertion (Supplementary Fig. 8c and ref. 43). We next monitored the Ssa1-to-Sgt2 TA transfer step using the Bpa crosslinking assay. Crosslink of Sgt2 with 4AG, 5AG, and 6AG were observed within 1 min after the addition of Sgt2 to the respective Ssa1-TA complexes, with efficiencies comparable to that of Bos1 (Fig. 7a), indicating that these TA variants were efficiently loaded onto Sgt2 by Ssa1. In contrast, no crosslink was detected between Fis1 and Sgt2 (Fig. 7b), and pulldowns on His$_6$-Sgt2 detected low amounts

of co-purified Fis1 (Supplementary Fig. 8d). These results suggest that Fis1 was not efficiently transferred to Sgt2 or that the Fis1 TMD did not properly engage the Sgt2 SBD.

Finally, we measured the kinetic stability of the complexes of Sgt2 with 4AG and 6AG using the FRET-based TA dissociation assay (Fig. 1a). While the kinetics of Bos1 dissociation from Sgt2 can be described by a single population with a slow rate (0.018 min$^{-1}$), the Sgt2-4AG complex showed an additional population (~30%) that dissociated 10-fold more quickly (Fig. 7c, d). 6AG dissociated rapidly from Sgt2 and displayed two populations with $k_{off}$ values ~10-fold and ~50-fold faster than that of Bos1 (Fig. 7c, d). Thus, suboptimal TAs can be rejected both during and after their transfer from Ssa1 to Sgt2.

We wondered whether the lower kinetic stabilities of the Sgt2-TA complexes observed with 4AG and 6AG stemmed from differences in the conformation of the Sgt2 dimer when it engages different

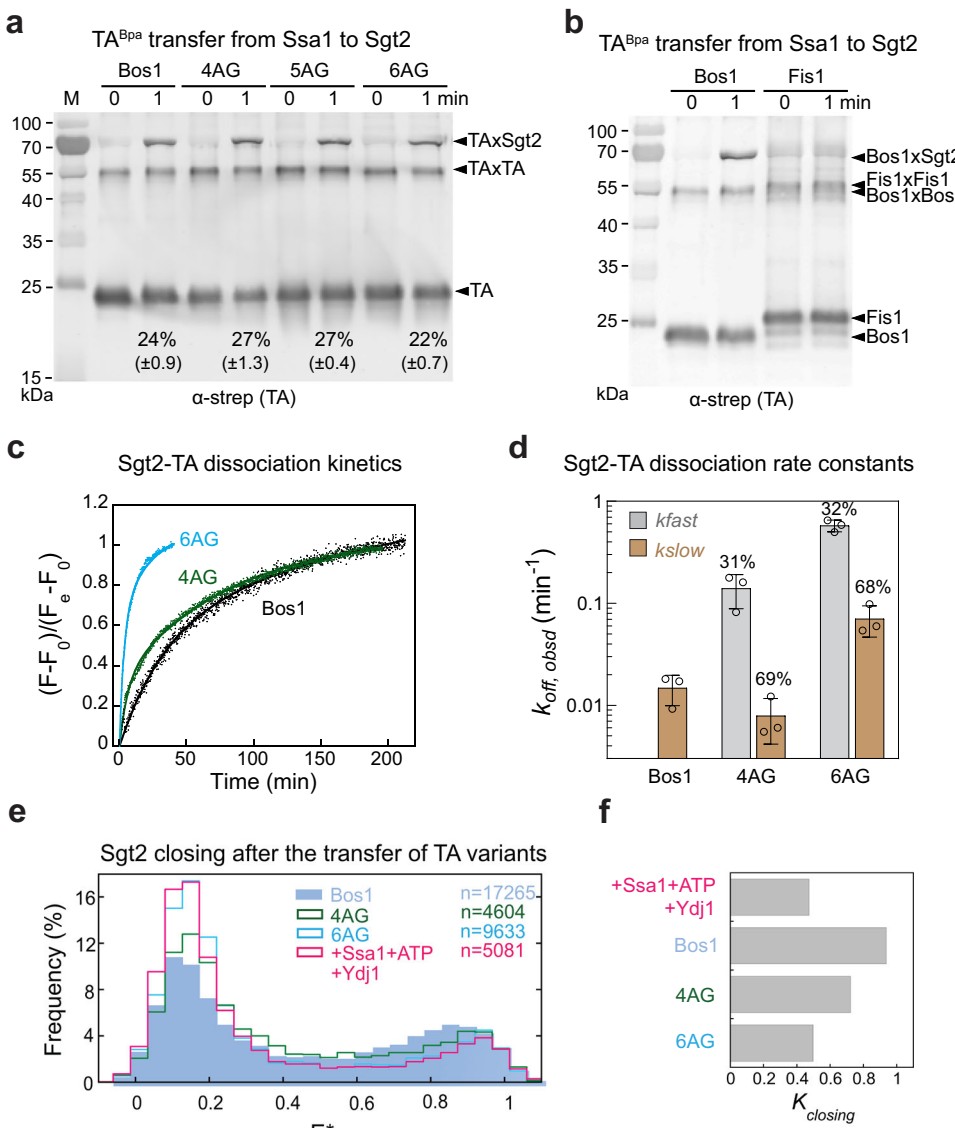

**Fig. 7 | Suboptimal TAs are bound to Sgt2 less stably, which correlates with reduced Sgt2 closing. a**, **b** 4AG, 5AG and 6AG were efficiently loaded onto Sgt2 by Ssa1 (**a**), whereas Fis1 was not (**b**). TA transfer was measured using UV-induced crosslink of TA$^{Bpa}$, as described in Methods. The percentage of crosslinked Sgt2-TA is indicated underneath and represented as mean ± SD, with $n$ = 3. **c**, **d** Suboptimal TAs dissociate more quickly from Sgt2. Time courses of the dissociation of Bos1, 4AG, and 6AG were measured as in Fig. 1c using cpSRP43 as chase (**c**). Data are normalized such that $F$ = 0 at $t$ = 0, and $F$ = 1 at the end of the reaction. The data for Bos1 were fit to a single exponential function (Eq. 1), and the data for 4AG and 6AG were fit to double exponential functions with a fast and slow phase (Eq. 2). The obtained rate constants are summarized in **d** as mean ± SD, with $n$ = 3. **e** smFRET histogram of Ssa1-Sgt2-TA for the indicated TA variants. The histogram for Sgt2 in the presence of Ssa1 and Ydj1 without TA (hot pink) is shown for comparison. **f** Summary of the equilibrium of Sgt2 closing obtained from mpH$^2$MM analysis. Source data are provided in the Source Data file.

substrates. We therefore carried out smFRET measurements on Ssa1-Sgt2-TA complexes formed with the suboptimal substrates. The equilibrium for Sgt2 closing after the Ssa1/Ydj1-mediated TA transfer reduced from 0.937 with Bos1 to 0.722 with 4AG, and 0.497 with 6AG (Fig. 7e, f; Table S1). The value of $K_{closing}$ after the transfer of 6AG was the same, within error, as that of Ssa1/Ydj1-bound Sgt2 without TA (Fig. 7f and Table S1). These results suggest that the extent of Sgt2 closing is fine-tuned by the biophysical properties of the TA-TMD, which could provide a mechanism to reject suboptimal TAs from the GET pathway.

## Discussion

Across all organisms, networks of molecular chaperones coordinate the capture and handover of newly synthesized MPs to their correct destinations in the cell. The GET pathway, in which nascent TAs are sequentially relayed through a cascade of chaperone (Hsp70), cochaperone (Sgt2), and targeting factor (Get3), provides a paradigm to understand the molecular mechanism underlying client handovers within a chaperone network and their roles in the efficiency and fidelity of MP targeting. The results of this work reveal fine-tuned conformational dynamics of Sgt2 that enable it to recognize, receive, and then handoff a hydrophobic TMD helix, and to do so for a diverse set of substrates while maintaining the specificity of the GET pathway.

Client interaction of Sgt2 is one of the least understood aspects of the GET pathway, with contradictory reports concluding that the Sgt2-TA complex is stable[18] or dynamic[40]. Here, kinetic analysis resolves this issue and reveals that the Sgt2-TA complex displays 'dynamic stability': the complex is kinetically stable, with a slow intrinsic TA dissociation rate, but can facilely hand off the bound TA to other Sgt2 molecules or to alternative chaperones such as CaM (Fig. 1). Single-molecule analyses suggested a molecular basis for this behavior: the Sgt2 dimer samples, on the millisecond timescale, open and closed conformations in which one or both of its SBDs engage the TA-TMD (Figs. 3–4). Rapid sampling of the closed state prevents TAs from dissociation, giving rise to the high kinetic stability of the Sgt2-TA complex (Figs. 1, 5). On the other hand, sampling of the open state, which exposes part of the TA-TMD, renders the Sgt2-TA complex susceptible to 'invasion' by other chaperones and to directly exchange the bound substrate with them (Figs. 1, 5). These exchanges could minimize the exposure of hydrophobic TMDs in the cytosol and their consequent misfolding and aggregation. Although additional examples of chaperone dynamic stability have not been reported, we speculate that this behavior could be beneficial in other protein biogenesis pathways that involve aggregation-prone client proteins, such as the interaction of nascent outer MPs with multiple periplasmic chaperones prior to their insertion at the bacterial outer membrane[53–55].

Importantly, Sgt2 closing is driven by upstream chaperones and the TA substrate, providing a mechanism for Sgt2 to sense and respond to the molecular signals that initiate TA targeting. ATP-bound Ssa1 induces the otherwise wide-open Sgt2 to rapidly sample the closed state (Fig. 4 and Table S1). The Ydj1-induced ATPase activation of Ssa1 further helps maximize Sgt2 closing (Fig. 4). Moreover, Ssa1 remains bound to Sgt2 after TA transfer and helps maintain Sgt2-TA in the closed conformation (Fig. 4). Together, these observations suggest that the Hsp40/Hsp70 chaperone cycle helps organize the first transfer complex and primes Sgt2 to capture the TA from Ssa1 in the optimal conformation. In support of this model, TA transfer from Ssa1 to Sgt2 is complete within 1 second, at least two orders of magnitude faster than TA dissociation from Ssa1, and is strongly protected from external TA-binding chaperones such as CaM (Fig. 6). This provides strong evidence for an active transfer mechanism in which the TA substrate is directly channeled from the SBD of Ssa1 to that of Sgt2 in the closed conformation. Notably, only Get3 can capture the bound TA from closed Sgt2 in the second transfer complex assembled by Get4/5 (Figs. 3d, e, and 5). This is distinct from off-pathway chaperones such as CaM,

which can only access TAs from open Sgt2, and provides a molecular explanation for why the Sgt2-to-Get3 TA transfer is also highly privileged. Collectively, these results demonstrate that GET-dependent substrates are rapidly channeled through the Ssa1-Sgt2-Get3 chaperone triad en route to their delivery to the ER.

The substrate channeling observed here is not limited to the GET pathway. Recent studies suggested that the glucocorticoid receptor (GR) is threaded through a channel formed by the HSP70-HOP-HSP90 complex during its folding[56,57]. Analogous to TA channeling in the Hsp70-Sgt2-Get3 chaperone triad, GR threading through the HSP70-HOP-HSP90 complex protects them from premature folding. Completion of GR folding requires exquisite molecular coordination between HSP70, HOP, and HSP90 to load the ligand binding domain of GR in the client binding pocket of Hsp90[56,57]. It can be envisioned that analogous coordination occurs between Ssa1, Sgt2, and Get3 during TA channeling. This could involve the accurate positioning of the SBDs of the upstream and downstream chaperones and possibly transient interactions between them. The architectural organization of the transfer complexes and the precise mechanism of molecular coordination that enables substrate channeling in this chaperone triad remain outstanding questions.

The abundance of Hsp70 in the cytosol[58], its ability to rapidly capture numerous newly synthesized proteins, including the TAs[59], and the rapidity of the Ssa1-to-Sgt2 TA transfer further suggest that the Hsp70-Sgt2 complex enjoys kinetic precedence over alternative TA targeting pathways, and proteins are re-routed to alternative biogenesis pathways only if they are rejected by this chaperone-cochaperone pair. Indeed, the conformation of Sgt2 is fine-tuned by the biophysical property of the TA-TMD, which could contribute to such rejection mechanisms. TAs delivered by the GET pathway are characterized by a higher hydrophobicity and helical propensity of their TMDs compared to TAs destined to mitochondria or delivered by alternative pathways to the ER[43,60,61]. Although TAs with suboptimal TMDs are still efficiently captured by Ssa1 and transferred to Sgt2, they are much less effective in inducing Sgt2 closing, and the more extensive sampling of the open state of Sgt2 correlated with the faster dissociation of these TA variants (Fig. 7). In addition, TA containing the TMD of Fis1, a mitochondrial protein, was inefficiently transferred to Sgt2 (Fig. 7). Thus, GET-independent TAs can be rejected both during and after their transfer from Ssa1 to Sgt2, in part by leveraging the open state of Sgt2. The ability of open Sgt2 to handoff the TA to other chaperones further suggests that suboptimal TAs could be directly re-routed to alternative biogenesis pathways without re-exposure of their TMDs to the cytosol.

Collectively, these results allow us to propose a revised model for the early molecular events during TA biogenesis. Ssa1 provides a 'hub' that captures diverse newly synthesized TAs. Dimeric Sgt2 by itself is predominantly in an open conformation in which the two SBDs are apart, but interaction with Ssa1 induces Sgt2 to sample the closed conformation in which its two SBDs are close together (Fig. 8, step 1). Rapid and protected TA transfer to Sgt2 ensues, and TAs with hydrophobic TMDs drive further closing of Sgt2 such that the TMD is stably bound and protected from off-pathway chaperones (step 2a). Sgt2 further recruits Get4/5, which positions Get3 to capture the TA from closed Sgt2 (step 2b). TA loading activates ATP hydrolysis on Get3 and drives its dissociation from Get4/5, committing TA for targeting to the Get1/2 receptor at the ER (steps 2c-2d). TAs with suboptimal TMDs can also be loaded on Sgt2 by Ssa1, but the Sgt2-TA complex more extensively samples the open conformation in which the TAs are less stably bound and can be handed off to alternative chaperones (steps 3a-3c). Additionally, mitochondrial TAs with even weaker TMDs can be rejected during the Ssa1-to-Sgt2 transfer and are likely delivered by Ssa1 to mitochondria (steps 4a–4b). We hypothesize that the Ssa1-Sgt2 pair comprises a decision point during the early stages of TA biogenesis. The fine-tuned conformational dynamics of Sgt2 enable this cochaperone to selectively funnel hydrophobic TAs into the GET

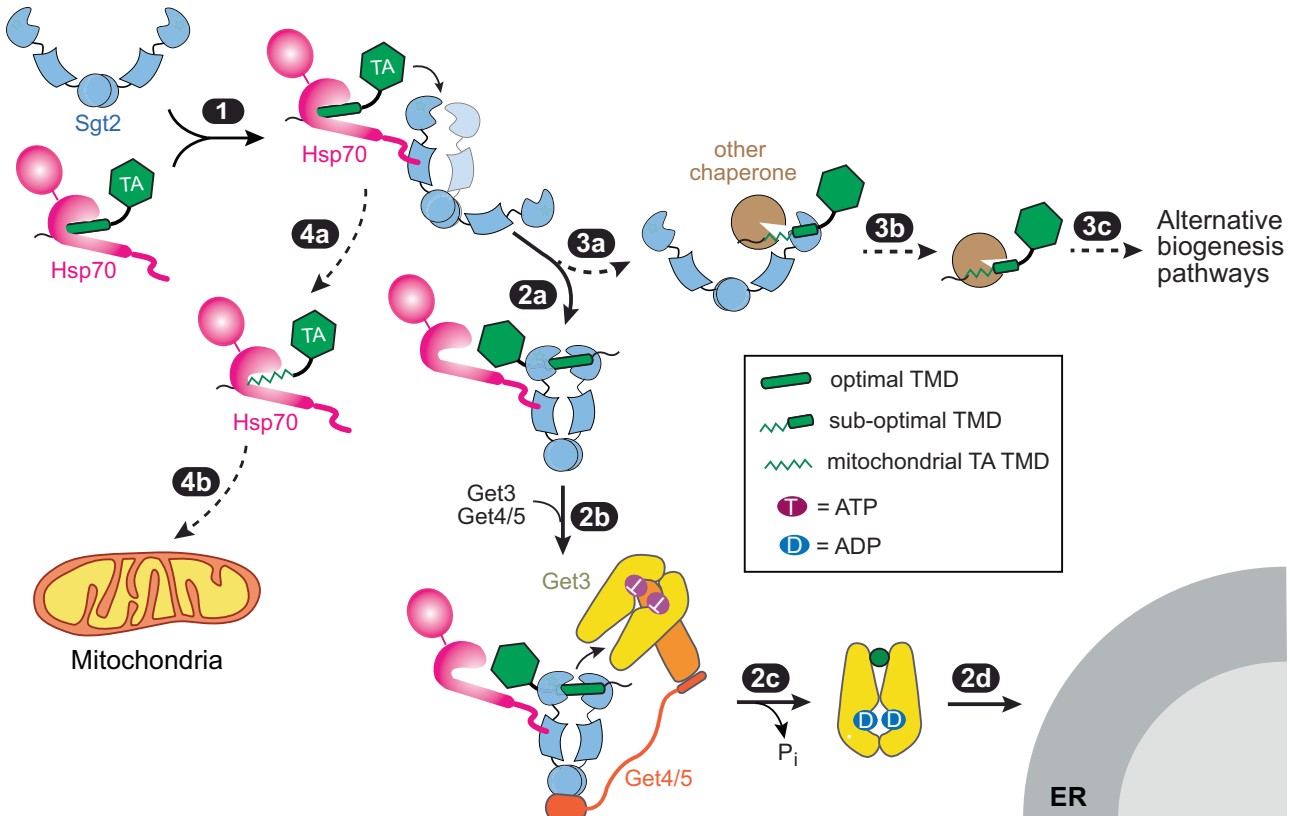

**Fig. 8 | Model for how the dynamic conformational changes of Sgt2 regulate TA targeting.** Interaction with Hsp70 induces wide-open Sgt2 to sample the closed conformation (step 1). TAs with hydrophobic TMDs rapidly transferred to Sgt2 and further favored Sgt2 closing, enabling tighter TA binding and its protection from off-pathway chaperones (step 2a). Get4/5-bound Get3 is uniquely privileged to receive the TA from closed Sgt2 (step 2b). TA loading activates ATP hydrolysis on Get3 and drives its dissociation from Get4/5 (step 2c), which commits Get3 to bind the Get1/2 receptors at the ER membrane (step 2d). Suboptimal TAs are also transferred to Sgt2 but are less effective in inducing Sgt2 closing (step 3a), allowing other chaperones to remove the TA from Sgt2 and re-route them to alternative pathways (steps 3b–3c). Mitochondrial TAs are also efficiently captured by Ssa1 but are rejected during the Ssa1-to-Sgt2 transfer (steps 4a-4b).

pathway, while also providing a mechanism to re-route alternative TAs and other hydrophobic proteins to their appropriate biogenesis pathways and cellular destinations.

## Methods

### Protein expression and purification

Model TA substrates for biochemical and biophysical studies were described previously[18,40]. In summary, Strep-SUMO-TA variants are composed of an N-terminal strep-tag, a non-cleavable SUMO domain, and a C-terminal targeting sequence that encompasses the TMD of Bos1 or its variants (4AG, 5AG, 6AG), Sbh1, or Fis1. Recombinant TA substrates were expressed, solubilized in LDAO (N,N-dimethyldode-cylamine N-oxide; Avanti Polar Lipids, Inc.), and purified using Strep-tatin resin (IBA Lifesciences) as described previously[18]. His$_6$-Sgt2, His$_6$-Sgt2ΔN (residues 96–346, a gift from Clemons lab), and His$_6$-Sgt2-LPATGG proteins were expressed and purified as described previously[43]. To minimize degradation of Sgt2, Sgt2-expressed cells were lysed using mild detergents (BugBuster, Merck)[43]. His$_6$-Get3[43], His$_6$-Get4/5[23], His$_6$-CaM[62] were expressed and purified as described previously. His$_6$-SUMO-Ssa1, His$_6$-SUMO-Ydj1, and His$_6$-SUMO-Ydj1-JD-GF(1–102aa) were expressed and purified as described[18,19]. Briefly, cells were resuspended in Buffer A [20 mM Tris (pH 8.0), 500 mM NaCl, 10% glycerol, 2 mM β-ME, 15 mM imidazole] supplemented with protease inhibitor cocktail. Clarified lysate was incubated with Ni Sepharose High-Performance resin (GE Healthcare) at 4 °C for 1 h. After washing the resin with Buffer A, the proteins were eluted from the column using Buffer A containing 300 mM imidazole. To obtain tagless Ssa1 and Ydj1, purified His$_6$-SUMO fusion proteins were digested with SUMO

protease overnight at 4 °C, and the proteins were further purified using MonoQ 5/50GL (GE Healthcare).

### Fluorescence labeling

Bos1 was labeled with CM (N-(7-Dimethylamino-4-Methylcoumarin-3-yl)Maleimide; Thermo Fisher Scientific Inc.) at a cysteine located at the seventh residue downstream of the TMD, as described[18]. Sgt2 was C-terminally labeled with BODIPY-FL (BFL) using Sortase-mediated ligation, as described[18,63]. Labeling of Sgt2 with Atto550 or Atto647N followed the same procedure as for BFL. Briefly, 30 μM His$_6$-Sgt2-LPATGG, 30 μM Sortase A, and 150 μM GGGC-Atto550 or 150 μM GGGC-Atto647N were incubated in Sortase Labeling buffer [50 mM Tris (pH 7.5), 150 mM NaCl, 10 mM CaCl$_2$] at 25 °C for 3 h. Labeled His$_6$-Sgt2 was purified using Talon resin (Clontech). To doubly label Sgt2 with both dyes, a mixture of 75 μM GGGC-Atto550 and 75 μM GGGC-Atto647N was used during the labeling reaction.

### In vitro translations

Sgt2-TA$^{Bpa}$, Sgt2-C258-TA$^{Bpa}$, and Sgt2ΔN-TA$^{Bpa}$ complexes were generated by IVT of Bos1 for 2 hrs at 30 °C using a home-made *E. coli* S30 lysate[43,45]. IVT reactions were supplemented with $^{35}$S-methionine, 2–5 μM His$_6$-tagged Sgt2-WT or Sgt2-G258C. Bpa was incorporated at an amber codon (TAG) located at position A228 in the Bos1-TMD by using bacteria expressing tRNA$_{CUA}$$^{opt}$ for preparation of the S30 extract and supplementing 10 μM purified $^{Bpa}$RS(D286R), the evolved tRNA synthetase for Bpa, during IVT[41,43,44]. For complexes that require subsequent BMH crosslinking, 1 mM TCEP (Gold Bio) was maintained throughout IVT and pulldown to ensure that the cysteine was reduced.

### Preparation of Sgt2-Ssa1-TA and Sgt2-TA complexes

Complexes for fluorescence measurements were generated by TA transfer from Ssa1 to Sgt2 using recombinant proteins followed by purification via His$_6$-tagged Sgt2, as outlined in Supplementary Fig. 1. 8 μM TA solubilized in TA buffer [50 mM K-HEPES pH 7.5, 300 mM NaCl, 20% Glycerol, 0.05% LDAO, 2 mM β-mercaptoethanol] was diluted to a final concentration of 0.3 μM in GET Buffer [50 mM HEPES (pH 7.5), 150 mM KOAc, 5 mM Mg(OAc)$_2$, 10% glycerol, 2 mM β-mercaptoethanol] containing 3 μM Ssa1, 2 mM ATP, and 3 μM Ydj1 where indicated. For smFRET measurements of the Ssa1-Sgt2-TA complex, 100–200 pM doubly labeled Sgt2$^{ATTO550/ATTO647N}$ was added to the preformed Ssa1-TA complex and incubated at room temperature for 1 min immediately before the measurement (Fig. 3b, 1st transfer). Where indicated, 0.5 μM Get3 with or without 0.5 μM Get4/5 was further added, and the mixture was incubated at RT for 10 min before the measurement (Fig. 3b, 2nd transfer).

To generate purified Ssa1-Sgt2-TA and Sgt2-TA complexes, 0.6 μM His$_6$-tagged Sgt2 was added to the preformed Ssa1-TA complex and incubated for 1 min. The reaction mixture was incubated with Talon resin (Clonetech) in GET Buffer at 4 °C for 10 min with agitation. To obtain the Ssa1-Sgt2-TA complex, the resin was washed in 75 column volume of GET buffer supplemented with 5 mM Imidazole and eluted with four-column volumes of GET buffer supplemented with 300 mM Imidazole. To purify the Sgt2-TA complex, the resin was sequentially washed with 100 column volumes of GET buffer supplemented with 500 mM NaCl and eluted with 4 column volumes of GET buffer supplemented with 300 mM Imidazole.

Sgt2-TA$^{Bpa}$ and Sgt2-C258-TA$^{Bpa}$ complexes were generated by IVT of Bos1$^{Bpa}$ in *E. coli* S30 lysate supplemented with $^{35}$S-methionine and 2–5 μM His$_6$-Sgt2 or His$_6$-Sgt2-G258C. Immediately after translation, Sgt2-TA complexes were purified via the His$_6$-tag on Sgt2 using Talon resin (Clonetech), as described in ref. 18, or via the Strep$_3$-tag on TA using Streptactin Sepharose resin (IBA LifeSciences), as described in ref. 43. The efficiency of TA translation and co-purification with Sgt2 was evaluated using SDS-PAGE and autoradiography. The concentration of Sgt2 in the purified complexes was determined using a Silver Stain kit (Pierce) or quantitative western blot, using known quantities of purified Sgt2 loaded on the same gel as a standard. For samples that require BMH crosslinking, 1 mM TCEP was present throughout IVT and purification to ensure that the cysteine on Sgt2-C258 was reduced. Purified Sgt2-C258-TA complex was incubated with 1 mM BMH (Thermo Fisher) for 1 h at RT to form the intradimer crosslink, and crosslinking was quenched with 10 mM DTT. Crosslinking efficiency was determined using immunoblot against anti-His primary antibody (Genscripts, Cat# A00186, 1:3000 dilution).

**TA transfer measurements.** For fluorescence-based assays, apparent rate constants of TA release from Sgt2 were measured by chasing 50–150 nM Sgt2$^{BFL}$-Bos1$^{CM}$ complex in GET buffer with 2.5–30 μM CaM or 14–28 μM cpSRP43. The time courses of donor fluorescence recovery (due to loss of FRET) were monitored using an excitation wavelength of 360 nm and emission wavelength of 460 nm on a Fluorolog-3-22 spectrofluorometer (HORIBA Scientific). The donor fluorescence intensity was normalized, and the data were fit to Eq. 1 or Eq. 2,

$$\text{Normalized fluorescence change} = \frac{F_{obsd} - F_0}{F_e - F_0} = A(1 - e^{-kt}) \quad (1)$$

$$\frac{F_{obsd} - F_0}{F_e - F_0} = A_{fast}(1 - e^{-k_{fast}t}) + A_{slow}(1 - e^{-k_{slow}t}) \quad (2)$$

in which $F_e$ is the donor fluorescence intensity when the reaction reaches equilibrium, $F_0$ is donor fluorescence intensity at $t = 0$, $k_{fast}$ and $k_{slow}$ are the rate constants of the fast and slow phases in the dissociation reaction, respectively, and $A_{fast}$ and $A_{slow}$ are the amplitudes

of the fast and slow phases, respectively. The only exception is Fig. 1f, in which $F_e$ is the donor fluorescence intensity when the reaction with 15 μM CaM plateaued.

For Bpa crosslinking-based assays to monitor TA transfer from Ssa1 to Sgt2, 0.1 μM recombinantly expressed and purified strep-tagged TA$^{Bpa}$ were preincubated with 3 μM Ssa1 in GET Buffer for 1 min at RT. 0.3 μM Sgt2 was added, and reactions were further incubated for 1 min before flash freezing in liquid nitrogen. 0 min samples were taken right before Sgt2 addition. Frozen reaction aliquots were crosslinked on dry ice -4 cm away from a UVP-B-100AP lamp (UVP LLC) for 20 min. Crosslinked and uncrosslinked TA were resolved on SDS-PAGE and visualized by anti-Strep primary (Genscripts, Cat# A01732, 1:3000 dilution) and IRDye® 800CW secondary antibodies (LiCor, Cat#926-32210, 1:15,000 dilution). The percentage of TA$^{Bpa}$ crosslinked to Sgt2 was quantified using Quantity One software (Bio-Rad).

TA transfer kinetics from Ssa1 to Sgt2 was measured using an RQF-3 quench-flow apparatus (KinTek). Briefly, 8 μM Ssa1 with or without 8 μM Ydj1 was premixed with 0.5 μM Bos1$^{Bpa}$ in GET buffer supplemented with 2 mM ATP and loaded into Syringe A. After 2 min incubation, the first push mixed the sample in Syringe A with an equal volume of GET buffer containing 2 mM ATP in Syringe B for 0.01 sec. A second push mixed the twofold diluted Ssa1-TA$^{Bpa}$ complex with an equal volume of 0.6 μM Sgt2 in GET buffer (in Syringe C). After the desired delay time, the reaction mixture was pushed into a pre-cooled 96-well plate in a dry ice-ethanol bath. The frozen samples were crosslinked and analyzed as described above.

For TA transfers with BMH-crosslinked Sgt2-C258, Bos1$^{Bpa}$ was translated and labeled with $^{35}$S-methionine in S30 lysate in the presence of Sgt2 or Sgt2-C258. The Sgt2-TA complexes were purified by pulldown of His$_6$-tagged Sgt2 and crosslinked by incubation with 1 mM BMH for 30 min. To monitor TA loss from Sgt2 to CaM, 15–20 μM CaM was added to 250–500 nM purified Sgt2-TA$^{Bpa}$ or BMH-crosslinked Sgt2-C258-TA$^{Bpa}$ in the presence of 1 mM Ca(OAc)$_2$, and the reaction was incubated at RT. To monitor the transfer of TA from Sgt2 to Get3, 0.75 μM Get3 was added to Sgt2- TA$^{Bpa}$ or BMH-crosslinked Sgt2-C258-TA$^{Bpa}$ in the presence of 2 mM ATP and 0.75 μM Get4/5 where indicated. At indicated times, aliquots were removed from the reaction and quenched by flash freezing in liquid nitrogen. Crosslinking was carried out ~4 cm away from the UVP-B-100AP lamp (UVP LLC) for 20 min on ice. Crosslinked and uncrosslinked $^{S35}$TA and Sgt2 were resolved on 9% Tris-Tricine gel and quantified by autoradiography using ImageQuant (Cytiva).

### Kinetic simulation

Kinetic simulations (Fig. 1d, e) were carried out using Berkeley Madonna v 9.1.3. The following equations were used to describe the spontaneous TA dissociation model in Fig. 1d:

$$\text{Sgt2} - \text{TA} \underset{k_{-1}}{\overset{k_1}{\rightleftharpoons}} \text{Sgt2} + \text{TA} \quad (3a)$$

$$\text{TA} + \text{CaM} \underset{k_{-2}}{\overset{k_2}{\rightleftharpoons}} \text{CaM} - \text{TA} \quad (3b)$$

$$\text{Normalized fluorescence} = 1 - \frac{[\text{Sgt2} - \text{TA}]}{[\text{Sgt2} - \text{TA}] + [\text{CaM} - \text{TA}] + [\text{TA}]} \quad (3c)$$

in which $k_1 = 0.015 \text{ min}^{-1}$ (as determined using cpSRP43 as chase), $k_{-1} = 1 \times 10^6 \text{ M}^{-1} \text{ min}^{-1}$, $k_2 = 10 \times 10^6 \text{ M}^{-1} \text{ min}^{-1}$, $k_{-2} = 10 \text{ min}^{-1}$.

The following equations were used describe the active model in Fig. 1e:

$$\text{Sgt2} - \text{TA} + \text{CaM} \underset{k_{-1}}{\overset{k_1}{\rightleftharpoons}} \text{Sgt2} - \text{TA} \cdot \text{CaM} \quad (4a)$$

$$Sgt2 - TA \cdot CaM \underset{k_{-2}}{\overset{k_2}{\rightleftharpoons}} Sgt2 + CaM - TA \quad (4b)$$

$$\text{Normalized fluorescence} = 1 - \frac{[Sgt2 - TA]}{[Sgt2 - TA] + [Sgt2 - TA - CaM] + [CaM - TA]} \quad (4c)$$

in which $k_1 = 0.035 \times 10^6\,M^{-1}\,min^{-1}$ (from the slope of the CaM chase data in Fig. 1g), $k_{-1} = 1\,min^{-1}$, $k_2 = 10\,min^{-1}$, $k_{-2} = 20 \times 10^6\,M^{-1}\,min^{-1}$. The overall equilibria of TA exchange between Sgt2-TA and CaM-TA in both Eqs. (1) and (2) were set to ~0.17, based on the observation that 2.5 μM CaM drove the dissociation of 40% of TA from 0.05 μM Sgt2 complex (Fig. 1f). The values of $k_{-1}$, $k_2$, and $k_{-2}$ were adjusted to be consistent with this overall equilibrium; otherwise, varying the specific values of $k_{-1}$, $k_2$ and $k_{-2}$ over a 10-fold range do not alter the kinetic pattern generated by the simulation. The initial concentrations used were: [Sgt2-TA] = 0.05 μM, [Sgt2] = 0; [TA] = 0; [CaM] as indicated in Fig. 1d, e; [Sgt2-TA•CaM] = 0. Equations 3c and 4c describe the fluorescence signal (in arbitrary units) based on the fact that FRET in the Sgt2-TA complex reduces donor fluorescence intensity; thus, donor fluorescence increase is proportional to loss of the Sgt2-TA complex during the measurement.

### Turbidity assay
TA variants solubilized in 0.05% LDAO were rapidly (within 15 s) diluted into GET Buffer (without glycerol) to a final concentration of 1.5 μM in the absence and presence of the indicated concentrations of Ssa1. The optical density at 360 nm was recorded using a spectrophotometer (Beckman Coulter). Observed % soluble TA ($S_{obsd}$) was calculated from the difference in optimal readings between reactions with and without Ssa1 5 min after initiation of the reaction and was normalized to that of Bos1 alone. The data were plotted as a function of Ssa1 concentration and fit to Eq. 5,

$$S_{obsd} = S_{Max} \times \frac{[Ssa1]}{K_{soluble} + [Ssa1]} \quad (5)$$

where $S_{Max}$ is the % soluble TA at saturating Ssa1 concentrations. $K_{soluble}$ is the concentration of Ssa1 required to solubilize 50% of the TA and is termed the apparent TA solubilization constant.

### μs-ALEX measurements
All protein samples were ultracentrifuged in a TLA100 rotor (Beckman Coulter) at $386,400 \times g$ for 30 min at 4 °C to remove aggregates before all measurements. All measurements were carried out in GET buffer supplemented with 0.3 mg/ml BSA. For each measurement, fluorescently labeled Sgt2$^{ATTO550/ATTO647N}$ or Sgt2$^{ATTO550/ATTO647N}$-TA complex was diluted to ~200 pM final concentration in GET buffer. Where indicated, reactions contained 2 mM ATP or ADP, 3 μM Ssa1, 3 μM Ydj1 or variants, and 0.5 μM Get3 and Get4/5. Samples were placed either on a coverslip (for measurements ≤15 min) or in a closed chamber made by sandwiching a perforated silicone sheet (Grace Bio-Labs) with two coverslips to prevent evaporation (for measurements >30 min). Data were collected using an ALEX fluorescence-aided molecule sorting setup[47] with two single-photon Avalanche photodiodes (PerkinElmer), 532-nm and 635-nm continuous wave lasers (Opto Engine LLC) operating at 135 μW and 80 μW, respectively.

### μs-ALEX data analysis
All μs-ALEX data analyses were performed using FRETBursts[64], a Python-based open-source burst analysis toolkit for diffusion-based smFRET. The burst considered to be the signal of a single molecule was determined as a minimum of 10 consecutive detected photons with a photon count rate at least 15-fold higher than the background rate

during both the donor and acceptor excitation periods. Due to the fluctuating background rate within a measurement, the background rate was computed for every 50-s interval according to maximum likelihood fitting of the interphoton delay distribution. Dual-channel burst search was performed to screen bursts of FRET pair species from bursts for mixtures containing donor or acceptor-only species. The identified bursts were further filtered according to the following criteria: (1) $n_D^D + n_D^A \geq 15$ (exclude acceptor-only species) and (2) $n_A^A \geq 15$ (exclude donor-only species), where $n_D^D$ is the number of photons detected from the donor channel during donor excitation, $n_D^A$ is the number of photons detected from the acceptor channel during donor excitation, and $n_A^A$ is the number of photons detected from the acceptor channel during acceptor excitation.

The values of the apparent FRET efficiency, $E^*$, and stoichiometry, $S$ were calculated for each burst according to the following equations:

$$E^* = \frac{n_D^A}{n_D^D + n_D^A} \quad (6)$$

$$S = \frac{n_D^D + n_D^A}{n_D^D + n_D^A + n_A^A} \quad (7)$$

The bursts with $S$ of ~1 and ~0 represent the donor-only and the acceptor-only species, respectively. The bursts with $S$ values between 0.2 and 0.8 are generally considered to be from doubly labeled FRET pairs. The accurate FRET efficiency can be calculated by applying three correction factors to $E^*$: (1) $\gamma$-factor, the ratio between quantum yields and detection efficiency between donor and acceptor channels; (2) $lk$-factor, the donor spectral leakage into the acceptor channel; and (3) $dir$-factor, the direct excitation of acceptor by donor excitation illumination. Those correction factors are expected to be constant as long as the same optical setup and FRET pair are used throughout all measurements. More importantly, no significant change in quantum yields of donor and acceptor was observed upon the binding of ligands and interaction partners (Supplementary Fig. 3c, d). Therefore, the trend of changes in $E^*$ due to the conformational changes of Sgt2 will match the trend of accurate FRET efficiency for each experimental condition. In this study, we used the uncorrected $E^*$ to generate all FRET histograms.

To qualitatively probe millisecond dynamics of Sgt2 under various conditions and binding partners, we performed burst variance analysis (BVA) as described in ref. 49. The probability that an acceptor emits a photon from a single molecule due to the FRET process can be described by a binomial distribution if there are no dynamic events in the single molecule. Therefore, the expected standard deviation (SD) of FRET due to the shot-noise limit for a burst with a given $E^*$ and $n$ photons is defined as the static limit and given by:

$$\sigma_{E^*} = \sqrt{\frac{E^*(1 - E^*)}{n}} \quad (8)$$

where $E^*$ is the apparent FRET efficiency of a burst, n is the number of photons emitted during the donor excitation period in a burst (i.e., $n_D^D + n_D^A$). The observed SD of $E^*$ for each burst was computed as follows:

$$SD\ of\ E^* = \sqrt{\frac{1}{M} \sum_{i=1}^{M} (e_i^* - E^*)^2} \quad (9)$$

where $E^*$ and $e^*$ are the apparent FRET efficiencies of a burst and a sub-burst (a subset of each burst containing a fixed number of consecutive photons), respectively, $M$ is the number of sub-bursts in a burst. The $n$ of 5 was used in this study.

To increase the statistical power and reduce possible errors due to the small number of sub-bursts in individual bursts, we binned bursts into 20 bins along the $E^*$ axis with a bin width of 0.05. The SD of $E^*$ for each bin ($SD_{E^*}$) was then computed using Eq. 10.

$$SD_{E^*} = \sqrt{\sum_{\substack{i\ where \\ L \le E_i^* < U}} \sum_{j=1}^{M_i} \left[ \frac{(e_{ij}^* - \mu)^2}{\sum M_i} \right]} \qquad (10)$$

where $\mu = \sum_{\substack{i\ where \\ L \le E_i^* < U}} \sum_{j=1}^{M_i} \left( \frac{e_{ij}^*}{\sum M_i} \right)$, $L$ and $U$ are the lower and upper bounds of the bin, $M_i$ is the number of sub-bursts in the $i$th burst, $e_{ij}^*$ is the proximity ratio of the $j$th sub-burst in the $i$th burst. To ensure that the $SD_{E^*}$ values for each bin are representative of the sample dynamics, the $SD_{E^*}$ values of bins containing at least 2.5% of the total bursts were considered and denoted as triangles in BVA plots in this study.

To estimate the rate constants of conformational dynamics of Sgt2, we performed multi-parameter (mp) Photon-by-photon Hidden Markov modeling (H²MM) analysis[50,51]. H²MM, developed by Pirchi et al., is a generalized hidden Markov Model (HMM) algorithm that uses photon arrival times as input to the analysis, enabling HMM machinery to be applied to diffusion-based smFRET experiments[51]. Recently, Harris et al. introduced mpH²MM by integrating an additional photon stream (for μs-ALEX, the acceptor excitation stream) into the conventional H²MM that only uses raw FRET efficiency. This multi-parameter approach improves the accuracy of H²MM and enables the distinction between FRET dynamics and dye photophysical events[51].

In the implementation of mpH²MM analysis for this study, we only tested models containing up to 5 states to reduce computational costs in finding optimized models and parameters for each sample. Harris et al. introduced Integrated Complete Likelihood (ICL), a reliable extremum-based statistical discriminator, to find the optimal state model for H²MM analysis of the system[50]. In this study, ICL was minimized for the 3-state model in the majority of cases, including apo-Sgt2. The 3-state model consists of two states for interconversion of Sgt2 between low FRET ($E^* \sim 0.2$) and high FRET ($E^* \sim 0.8$) states, and a third state for dye photophysics (e.g., acceptor photoblinking). In some cases, alternative state models minimized the ICL; however, all H²MM analysis results of the alternative state models showed one or more states with transition rates that were too slow ($k \ll 1\,s^{-1}$) to detect using H²MM analysis and hence physically unreasonable (Supplementary Data 1). Based on these observations and the fact that the number of conformational states in Sgt2 is an intrinsic property of the molecule that cannot be altered by ligands and binding partners, the 3-state model was employed to perform H²MM analysis on all Sgt2 complexes used in this study.

After Viterbi analysis (in H²MM analysis), successive photons classified as belonging to the same state are grouped into dwells. The photons in the dwells can then be used to calculate $E^*$ and $S$ values for each dwell using Eqs. 11 and 12. These equations are essentially the same as for bursts described in Equations 6 and 7.

$$E_{Dwell}^* = \frac{n_{D,Dwell}^A}{n_{D,Dwell}^D + n_{D,Dwell}^A} \qquad (11)$$

$$S_{Dwell} = \frac{n_{D,Dwell}^D + n_{D,Dwell}^A}{n_{D,Dwell}^D + n_{D,Dwell}^A + n_{A,Dwell}^A} \qquad (12)$$

The equilibrium for the open-to-closed conformational change of Sgt2 was calculated using Eq. 13, based on the rate constants for the interconversion of open and closed states in the 3-state model

extracted from the H2MM analysis.

$$K_{closing} = \frac{k_{open \to closed}}{k_{closed \to open}} \qquad (13)$$

The uncertainty in the interconversion kinetic constants was estimated by (1) dividing the total bursts into subsets that contain 1700-2000 bursts, (2) performing mpH²MM analysis on the subsets with the 3-state model to extract kinetic constants for each subset, and (3) calculating the standard deviation of the kinetic constants. The uncertainty in the equilibrium constants ($K_{closing}$) for each sample was obtained by error propagation over the uncertainties in kinetic constants according to Eq. 13.

### Reporting summary

Further information on research design is available in the Nature Portfolio Reporting Summary linked to this article.

## Data availability

All data generated in this study are provided in the manuscript's main text, Supplemental information, Supplementary Data 1, and Source Data file. A previously published structure of E.coli DnaK used to design fluorescent dye labeling positions is available under PDB code 4EZO. Source data are provided with this paper.

## Code availability

All mpH2MM analyses in this study were performed using the Python packages, burstH2MM (Available at https://github.com/harripd/burstH2MM) and H2MM_C (Available at https://github.com/harripd/H2MMpythonlib).

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

## Acknowledgements
We thank members of the Shan laboratory for comments on the manuscript and A. Siegel for assistance with the quantification of Bpa crosslinking data. This work was supported by National Institutes of Health grants R35 GM136321 to S.-o. Shan and by Dean Willard Chair funds to S.W.

## Author contributions
H.C., Y.L., S.S. designed research; H.C., Y.L. and S. Chandrasekar performed biochemical experiments and analyzed data; Y.L. and S. Chung performed μs-ALEX experiments and analyzed data; S.W. provided guidance for μs-ALEX analysis; H.C., Y.L. and S.S. wrote the manuscript with input from S. Chandrasekar, S. Chung, and S.W.

## Competing interests
The authors declare no competing interests.
