## [Peer Review File · Nature Communications]

Reviewers' Comments:

Reviewer #1:

Remarks to the Author:

Cho et al investigate the targeting of proteins to membranes. In particular they investigate the conformational dynamics of Sgt2 and how this dynamic is crucial for targeting of proteins to the endoplasmic reticulum via the guided-entry-of-tail-anchored protein (GET) pathway. The authors claim that dynamic and fine-tuned conformational rearrangements in Sgt2 enable GET substrates to be effectively funneled onto their dedicated targeting factor. In addition, they providing a mechanism on how suboptimal substrates could be re-routed to alternative biogenesis pathways. Targeting of proteins and protein handover in the GET pathway is a very important topic. The authors' findings would significantly enhance our understanding and contribute new insights to the established literature.

Unfortunately, in my opinion, some claims are unclear and/or not supported by the provided data. I cannot support publication of the manuscript before the following points have been addressed:

- 1) Fig. 1B,C,F, 6C, 7C: What exactly is shown here? I assume it is the normalized change in donor fluorescence (what is it normalized to?). Please also show the acceptor fluorescence for these figures. The power of FRET is that both signals (donor and acceptor) are available and both together are important to exclude artifacts and judge the quality of the data.
- 2) Fig. 1B,C,F, 6C, 7C: How many independent repeats were done (different protein purifications)? Also for Fig. 3 and Fig. 4: How many independent repeats? This is very important, as the differences are often small.
- 3) Fig. 3E, 4D: How was 'open' and 'closed' distinguished - just a threshold in FRET efficiency? Please specify.
- 4) Figure S4: What is the scale shown on the right (blue colors)? If this is the number of observables, I cannot see a significant deviation from the dashed lines. Can this be quantified? Is there any possibility to calculate a significance?
- 5) The authors perform HMM to obtain kinetic rate constants. Please specify the model used in this HMM and how this model is justified.
- 6) Please explain the evidence for the several protein transfers claimed in this manuscript. In my view the fluorescence data shows that, e.g., in the presence of Ssa1 Sgt2 shows more closed conformation and upon the addition of Get3 a more open conformation. This is not an evidence for a transfer of protein. Only thermodynamic equilibrium states are reported, which do not provide information on pathways (succession of states).
- 7) If I understand correctly, Fig. 5D-G indicate that Sgt2-to-Get3 TA transfer occurs directly - not via dissociation and rebinding. Then spontaneous TA dissociation from Sgt2 should be much slower than this transfer - is this the case? Later the authors argue that 'the Ssa1-to-Sgt2 TA transfer does not occur via the dissociation of TA from Ssa1 followed by rebinding to Sgt2'. Here only indirect evidence is provided - could you please show similar time-dependent TA crosslink experiments as for Sgt2-to-Get3?
- 8) Fig. 7E,F is used to argue for a fine-tuning by biophysical properties. In my view, some things need to be clarified here: i) The changes in populations look very small. What would be the batch-to-batch variation of such an experiment? ii) Only thermodynamic equilibrium states are shown, which do not provide a mechanism. How is the mechanistic insight justified? iii) What are the 'biophysical properties' here?

Minor points:

- 9) Fig. 1H is, as far as I can see, pure speculation - please indicate this in the figure caption, or explain the evidence.
- 10) What is 'high intrinsic kinetic stability'? Please explain or reword.
- 11) What is 'dynamic stability'? As far as I understand, what the authors show is displacement of proteins in a competitive way, i.e. nothing that requires a new term.

Reviewer #2:

Remarks to the Author:

I'm grateful for the opportunity to review this interesting paper which casts light on TA-Sgt2

interactions through an elegant, extensive and cohesive experimental program. This work will certainly be a valuable contribution to the current understanding of proteostasis. I include a few points for clarification below:

Towards the end of page 3 the paper states: 'Biochemical, structural, and yeast genetics studies showed that the major cytosolic Hsp70 (Ssa1 in yeast), with the help of Hsp40 (Ydj1 or Sis1 in yeast), effectively captures newly synthesized GET substrates and is important for maintaining them in a soluble, targeting-competent conformation^{17–19}. Hsp70 further initiates a sequential series of energetically downhill TA transfers.'

This is not the only interpretation of events surrounding the initial capture of TAs. Several studies indicate that Get4/5 complex binds directly to ribosomes in yeast (Ribosome-bound Get4/5 facilitates the capture of tail-anchored proteins by Sgt2 in yeast

<https://pubmed.ncbi.nlm.nih.gov/33542241/> & Cotranslational Intersection between the SRP and GET Targeting Pathways to the Endoplasmic Reticulum of *Saccharomyces cerevisiae*

<https://www.ncbi.nlm.nih.gov/pmc/articles/PMC5007794/>) and in the equivalent mammalian system SGTA is reported to have direct interaction with nascent TAs in the ribosomal tunnel even when it bears mutations that negate its ability to bind to Hsp70/40 (SGTA associates with nascent membrane protein precursors <https://pubmed.ncbi.nlm.nih.gov/32216016/>).

The results presented in the current manuscript also point towards flexibility in the order of things e.g. from page 10: '...the TA substrate alone increased the population of closed Sgt2 from 18% to 38% (Fig. 4A and 4D, dark green; Table S1). This indicates that the TA substrate can by itself drive substantial Sgt2 closing, and reciprocally, closed Sgt2 binds TA with higher affinity than open Sgt2.' The ambiguity (and likely flexibility) of this situation should be explained with references included to support all available perspectives, especially considering this is rightly done in the case of other controversial aspects e.g. the substrate interaction kinetics.

Further, in the middle of page 4: '...raising questions about whether both SBDs in the Sgt2 dimer are required for TA binding. However, previous SAXS (small-angle X-ray scattering) and NMR analyses suggested that the Sgt2 dimer adopts a wide-open conformation in which the two SBDs are predicted to be 130 Å apart...' This is true but neglects to mention a highly pertinent paper in which full-length SGTA has been shown to display open and closed conformations relating to its substrate interactions (Structural complexity of the co-chaperone SGTA: a conserved C-terminal region is implicated in dimerization and substrate quality control

<https://pubmed.ncbi.nlm.nih.gov/29996828/>). There are two other important recent papers on the C-terminal domain omitted from this account which should be discussed either in the introduction or discussion to give a complete picture about what is currently known of this elusive domain (Structural insights into metazoan pretargeting GET complexes

<https://pubmed.ncbi.nlm.nih.gov/34887561/> Insights into the structure and function of the C-terminus of SGTs (small glutamine-rich TPR-containing proteins): A study of the *Aedes aegypti* homolog <https://pubmed.ncbi.nlm.nih.gov/34082040/>).

The methods need more details. On page 18 the protein production for a number of different proteins is listed as 'described previously' with a large selection of papers cited and no indication of which reference refers to which protein. If someone were to wish to reproduce specific proteins from this study they would be sent down a rabbit hole of research and even then could not be sure if they were finding the correct protocol. To be able to compare this study accurately with others in the literature it is very important to know how these (some of them potentially challenging) proteins were prepared – particularly things like whether reducing conditions and/or detergents were used at any stage. Regarding the construct Sgt2ΔN, I can't tell where the construct starts in the sequence and I also can't see evidence (apologies if I've missed this) that the protein remains stable throughout the experiments. Maybe it is fine but if it aggregated or degraded that would also account for the TA binding result. It seems plausible that, as suggested, the 2 C-terminal domains would need to be secured by the obligate N-terminal dimerization domains for efficient grasping of both sides of a TA but efficient binding between the isolated C-terminal domain and TAs is well documented (most recently in Molecular basis of tail-anchored integral membrane protein recognition by the cochaperone Sgt2 <https://pubmed.ncbi.nlm.nih.gov/33610544/> which is referenced elsewhere in the manuscript) – maybe the attached TPRs flopping around impede the TA binding of the Sgt2ΔN construct but this should be discussed.

In the initial chase experiments on page 6, the authors find that unlabeled Sgt2 can displace TA binding at a similar rate to CaM which makes me wonder whether the equivalent competition was tried in the ALEX experiments (in which Sgt2 was differentially FRET labeled on either side of the dimer - page 9) to see whether adding unlabeled Sgt2 would similarly displace the TA and veer the labeled Sgt2 back towards favoring the open conformation.

I must state that I am not experienced with many of the techniques used in this paper but the authors explain them admirably well and the interpretations of results seem justified. I would say that that sometimes more caveated language should be used as the effects, especially with regard to the different TA variants that veer towards hydrophobicity are arguable somewhat subtle.

I'm not sure about the word privilege - it occurs 15 times in the manuscript - instinctively it doesn't come across as quite the right word - but maybe I'm just unfamiliar with this usage and I'm very much willing to concede to the opinions of other reviewers on this. I think I get what the authors mean - that the system prioritizes more hydrophobic TAs for directional targeting to the GET pathway but privilege suggests something like positive discrimination to me whereas I envisage Sgt2 to be sorting substrates and sending them to different fates, rather than preferentially picking the ones that exceed a hydrophobicity threshold - maybe this is more true of SGTA though which associates with more complex machinery than the yeast system.

Finally a small formatting thing - for lists of sequences e.g. in Fig S8A it's much easier to look at if Courier font is used so that each letter takes up the same amount of space and the alignment is clearer.

Much of the confusion revolves about previous literature on the GET system, which we clarified below, and on the single molecule data analysis, which was previously buried in SI methods and which we have now moved to more prominent places. We also collected additional smFRET data on some of the samples to give more confidence in the role of the Sgt2 conformational dynamics in substrate specificity. Below are our point-by-point responses (blue) to these comments (black).

Reviewer #1 (Remarks to the Author):

Cho et al investigate the targeting of proteins to membranes. In particular they investigate the conformational dynamics of Sgt2 and how this dynamic is crucial for targeting of proteins to the endoplasmic reticulum via the guided-entry-of-tail-anchored protein (GET) pathway. The authors claim that dynamic and fine-tuned conformational rearrangements in Sgt2 enable GET substrates to be effectively funneled onto their dedicated targeting factor. In addition, they providing a mechanism on how suboptimal substrates could be re-routed to alternative biogenesis pathways.

Targeting of proteins and protein handover in the GET pathway is a very important topic. The authors' findings would significantly enhance our understanding and contribute new insights to the established literature.

Unfortunately, in my opinion, some claims are unclear and/or not supported by the provided data. I cannot support publication of the manuscript before the following points have been addressed:

1) Fig. 1B,C,F, 6C, 7C: What exactly is shown here? I assume it is the normalized change in donor fluorescence (what is it normalized to?). Please also show the acceptor fluorescence for these figures. The power of FRET is that both signals (donor and acceptor) are available and both together are important to exclude artifacts and judge the quality of the data.

$F = 1$ at the end of the time trace. The only exception is Figure 1F, where the fluorescence end point for the reaction with 15 μM CaM was set as 1. We have explained this in the figure legend.

The Sgt2-TA FRET assay was established in Cho et al, EMBO J. 2018 and that data are shown below. FRET is real because addition of unlabeled Sgt2 does not change fluorescence (i.e., no environmental sensitivity), and because the acceptor-induced reduction in donor fluorescence can be restored by unlabeled Sgt2 (i.e, reversibility). The same controls are shown for the Ssa1-TA FRET pair in Figure 6B.

Donor fluorescence intensity is monitored in a FRET measurement because this is how FRET is defined: $E = 1 - F_{DA}/F_D$ (Lakowicz, Principle in Fluorescence Spectroscopy, 2nd ed, Chapter 13, p369-370). It is standard practice in spectroscopic FRET measurements to keep the donor labeled molecules in trace amounts and acceptor-labeled molecules in excess. Monitoring both donor and acceptor fluorescence is only viable for experiments where both interaction partners are in stoichiometric amounts, but this is untenable in many cases. In this work, TA concentration was kept at <50 nM due to their high aggregation propensity. In addition, the K_d for TA of Ssa1 is $\sim 0.5 \mu\text{M}$. To ensure that all TAs are bound, Ssa1 were at 0.5-3 μM in our measurements, in >20-fold excess over TA. This means that only a small fraction of the chaperone would be TA-bound and FRET with TA. Therefore, the percentage change in acceptor fluorescence signal is small and is not meaningful to follow.

2) Fig. 1B,C,F, 6C, 7C: How many independent repeats were done (different protein purifications)? Also for Fig. 3 and Fig. 4: How many independent repeats? This is very important, as the differences are often small.

TA transfer rate from Sgt2 to Get3 reported in Fig. 1B and Fig. 1C was well characterized in our previous studies (Rao et al, eLife 2016, Cho et al, EMBO 2018), and one of the published data was used for comparison (we have edited the figure legend). TA dissociation from Sgt2 with the cpSRP43 chase had four replicates, and two representative data were shown in Fig. 1B and Fig. 1C. In addition, these rates are comparable to those in our previous study (Cho et al, EMBO 2018).

TA dissociation with the CaM chaperone in Fig. 1B and with the Sgt2 chaperone in Fig. 1C have duplicates, and the other data set, which largely overlaps with those in Figs. 1B & 1C, is shown below (Figure 2A-response and Figure 2B-response, respectively). In addition, rapid TA dissociation by CaM is observed independently using Bpa crosslinking, both previously (Shao et al, Science 2017; Chio et al, Cell Reports 2019) and in this work (Fig. 5B).

Figure 2-response. Additional replicates of TA dissociation from Sgt2 with different chaperones.

The data in Figs. 1F, 1G and 6C, 6D were measured in duplicates, and one of the datasets is shown in Fig. 1F and Fig. 6C. For Fig. 1G, we have added the extracted rate constants from the second data set, which are very similar to those from the other dataset. For Fig. 6D, the extracted rate constants from both datasets were summarized with statistics (mean \pm SEM). The data in Fig. 7C & 7D have 3 replicates, which is indicated in the figure legend.

For single molecule (sm) experiments, we added the # of replicates in SI Table 1. The smFRET histograms in the figures are based on the collection of all replicates specified in SI Table 1. The number of data points used to construct the smFRET histogram ('n') was specified for every sample in the histograms. Error estimates for the rate and equilibrium constants from the mpH²MM analysis are also provided in SI Table 1.

3) Fig. 3E, 4D: How was 'open' and 'closed' distinguished - just a threshold in FRET efficiency? Please specify.

Because BVA analysis showed that FRET bursts with intermediate E values are not real states, but rather a result of exchange averaging between conformational states, we performed multi-parameter (mp) photon-by-photon hidden Markov Analysis (H²MM) to the smFRET data to extract the true state parameters (conformational states, the mean E value of each state, interconversion rates). We added more explanation about the mpH²MM in the Results (p 9-10) and moved representative figures showing the results of the mpH²MM analysis to Figure 3. The results of mpH²MM analyses for all the other samples are in Appendix 1.

4) Figure S4: What is the scale shown on the right (blue colors)? If this is the number of observables, I cannot see a significant deviation from the dashed lines. Can this be quantified? Is there any possibility to calculate a significance? The color bars on the right of BVA plots indicate the number of observables corresponding to the color gradient. As shown in Fig. S4, the intermediate E* region deviated from the static limit (i.e., dashed line). Although this region has a low number of observables, this is due to the slow (on the order of milliseconds) dynamic exchange between the low (open conformation) and high (open conformation) E* states and does not indicate the presence of a small number of dynamic molecules. The figure below shows the result of a simple simulation (with no background and no dye photophysics) for the molecules that interconvert between two states every 5 milliseconds ($k_1=k_{-1}=200\text{ s}^{-1}$). The simulated BVA plot for this model (right) is similar to the observed BVA plots in Fig S4 and is consistent with Sgt2 dynamics occurring on timescales of 2~10 milliseconds according to mpH²MM analysis.

The FRET histogram of the simulated molecules (left bottom) shows two Gaussian-like distribution connected via a bridge in the intermediate E* region, which is also consistent with the observed histograms for Sgt2. Due to the slower transition compared to the molecular diffusion through the detection volume (~1 ms), molecular dynamic signatures are less frequently captured in the middle E* region. However, the observed dynamic signature clearly shows the deviation from the static limit in the BVA plot (please see the triangles plotted as results of variance analysis for the binned bursts). An increase in interconversion kinetics increases the number of observables in the middle E* region as the probability of capturing transitions increases.

5) The authors perform HMM to obtain kinetic rate constants. Please specify the model used in this HMM and how this model is justified.

The mpH²MM analysis of all the samples was performed using a three state model with two true conformational states (low E and high E with $S \sim 0.5$) and one subpopulation that arise from dye photophysics such as acceptor blinking ($E \sim 0$ and $S \sim 1$) (Figure 3E and Appendix 1). This information was in the SI Methods section, which we now moved to the Methods in the main text. We have also added a paragraph in the Results to explain this (p 9-10). In most samples, this three-state model is minimized in terms of integrated complete likelihood (ICL, described in ref. 50) and is therefore the most likely model. In a few cases, the ICL is slightly lower for a four-state or five-state model, but these models are excluded because the extracted interconversion rates are physically unreasonable (Appendix 1). For consistency, all the interconversion rates and state equilibrium reported here are based on the 3-state model.

6) Please explain the evidence for the several protein transfers claimed in this manuscript. In my view the fluorescence data shows that, e.g., in the presence of Ssa1 Sgt2 shows more closed conformation and upon the addition of Get3 a more open conformation. This is not an evidence for a transfer of protein. Only thermodynamic equilibrium states are reported, which do not provide information on pathways (succession of states).

Both transfers in the pathway were established in a series of previous studies. Get4/5-dependent Sgt2-to-Get3 TA transfer was shown by Wang et al, Mol Cell 2013 and corroborated in multiple studies by other groups (Shao et al, Science 2017; Chio et al, CellRep 2019; Rao et al, eLife 2016; ref 35, 36, 38). TA transfer from Ssa1 to Sgt2 was shown in Cho et al, EMBO J 2018 and corroborated in Cho et al, JBC 2021 (refs 18-19). Both transfers are shown based on the following: (i) transfer of the Bpa crosslinking of TA from the upstream to the downstream chaperone, analogous to the data in Figure 5 in this manuscript; (ii) FRET between TA and the downstream chaperone when the latter is presented to a preformed complex of TA with the upstream chaperone; (iii) Mutations that disrupt the contact between upstream and downstream chaperones strongly reduce TA loading on the downstream chaperone in both Bpa crosslinking and FRET-based assays in vitro and in co-IP experiments in vivo. (iv) absence of back transfer of TA from the downstream to the upstream chaperone, indicating directionality; (v) in the case of the Ssa1-to-Sgt2 transfer, preloading the TA on Ssa1 and then transfer to Sgt2 is essential for generating soluble, targeting competent Sgt2-TA complexes, whereas direct TA loading on Sgt2 generated mostly aggregated material, arguing for an ordered multistep pathway for productive TA loading on Sgt2; (vi) as shown in this manuscript, the rate constant of the transfers are $>10^2$ -fold faster than that of spontaneous TA dissociation from the upstream chaperone, arguing against simple competition between the chaperones.

Much of the manuscript tries to understand the nature of the TA transfers and the role of the Sgt2 conformational dynamics in the transfer, but is not meant to re-establish these transfer events. However, Figure 5 and Figure 7A used crosslinking of TA^{Bpa} with different chaperones to directly visualize and re-affirm many of these transfers.

7) If I understand correctly, Fig. 5D-G indicate that Sgt2-to-Get3 TA transfer occurs directly - not via dissociation and rebinding. Then spontaneous TA dissociation from Sgt2 should be much slower than this transfer - is this the case? Later the authors argue that 'the Ssa1-to-Sgt2 TA transfer does not occur via the dissociation of TA from Ssa1 followed by rebinding to Sgt2'. Here only indirect evidence is provided - could you please show similar time-dependent TA crosslink experiments as for Sgt2-to-Get3?

The entire Figure 1 was dedicated to measuring spontaneous TA dissociation rate, which was controversial in the field. This study now unequivocally establishes that spontaneous Sgt2-TA dissociation occurs at 0.015 min^{-1} (i.e., $\tau \sim 40 \text{ min}$). In contrast, TA transfer to Get3 is complete within 20 sec (Fig. 1C, orange; Fig. 5D-G).

Crosslinking-based TA transfer from Ssa1 to Sgt2 are shown in Figures 6E and 6F, and in previous work (Cho et al, EMBO J 2018; JBC 2021). Note that TA crosslink to Ssa1 is weak and diffuse on the gel, and usually not visualized well. However, direct TA loading on Sgt2 without Ssa1 is inefficient, and disrupting the Sgt2-Ssa1 contact in the Sgt2 TPR mutant (TPRmt) also disrupts TA loading on Sgt2; these results provide strong evidence of TA transfer.

8) Fig. 7E,F is used to argue for a fine-tuning by biophysical properties. In my view, some things need to be clarified here: i) The changes in populations look very small. What would be the batch-to-batch variation of such an experiment? ii) Only thermodynamic equilibrium states are shown, which do not provide a mechanism. How is the mechanistic insight justified? iii) What are the 'biophysical properties' here?

We repeated the measurement with 6AG with reprepared proteins, as closer inspection of the previous measurement suggested the presence of small aggregates. The new sample was well behaved, and we have made 9 new replicates of this measurement. mpH²MM of the new data show that the Sgt2 closing equilibrium is 0.49 with 6AG, which is largely the same as the same sample without TA (+Ssa1+ATP+Ydj1; $K_{closing} = 0.47$).

At this point in the manuscript, the collective data in Figures 1-6 has established that closed Sgt2 binds TA tightly, protects TA from off-pathway chaperones, and selectively transfers TA to Get3 for ER targeting, whereas open Sgt2 binds TA less tightly and can easily lose the bound TA to off-pathway chaperones such as CaM. In Figure 7, we asked whether this mechanism contributes to the sorting of different TAs, by testing if changing the features of the TA-TMD alters the distribution of Sgt2 between open vs closed states. If a weaker TA is less effective in inducing Sgt2 closing, they will be less likely to commit to the GET pathway and more likely to be re-routed to other chaperones/pathways. This prediction was tested with the TA variants and supported by the data.

We are confused by the reviewers' assertion that "only thermodynamic equilibrium states are shown". Figures 1, 5, and 6 are all kinetic measurements. The mpH²MM analysis showed that Sgt2 conformational transitions occur on the millisecond timescale and rates are reported (Table S1). We did not emphasize these exchange rates in the main text because they are much faster than the TA transfers (milliseconds vs seconds). Therefore, the equilibrium distribution of Sgt2 in the open vs closed states will dictate whether a TA commits to Get3 for ER targeting or gets re-routed.

"Biophysical properties" refer to the hydrophobicity (GRAVY scores) and helical (Agadir scores) propensities of the TA-TMD. We have now listed these scores in Figure S8A.

Minor points:

9) Fig. 1H is, as far as I can see, pure speculation – please indicate this in the figure caption, or explain the evidence. We rephrased in the text that this is a hypothesis. Different aspects of this hypothesis are then tested in the subsequent experiments. Figure 2 shows that both SBDs of Sgt2 are required for stable TA binding. Figures 3-4 showed that Sgt2 exchanges between open and closed conformations. Figure 5 showed that CaM accesses Sgt2-bound TA only in the open state.

10) What is 'high intrinsic kinetic stability'? Please explain or reword.

Slow k_{off} = high kinetic stability, the latter is a well-accepted term to convey the former.

In the case of Sgt2, we need to add "intrinsic" to distinguish spontaneous dissociation from chaperone-induced TA loss that occurs through a different pathway and occurs at much faster rates than spontaneous TA dissociation. This distinction is especially necessary given that a previous study has concluded that Sgt2 rapidly binds and releases TA based on observation of CaM-induced TA loss from Sgt2, which we now show is accelerated.

11) What is 'dynamic stability'? As far as I understand, what the authors show is displacement of proteins in a competitive way, i.e. nothing that requires a new term.

'Competition' typically refers to two chaperones binding free TA but not their complex with another chaperone; this model is depicted in Fig. 1D. In this mechanism, increasing concentration of the competing chaperone will only change the equilibrium but not the rate of TA dissociation from Sgt2. This is shown in the kinetic modeling in Figure 1D. In contrast, we observed that the rate of TA loss from Sgt2 is strongly accelerated by increasing concentrations of the chase chaperone. This kinetic effect can only be explained by a different pathway in which the transition state involves interaction of the chase chaperone with the Sgt2-TA complex. This is shown by the kinetic modeling in Figure 1E.

We defined 'dynamic stability' at the end of Section 1. It refers to chaperone-client complexes that have slow spontaneous substrate dissociation rates, but can be 'invaded' by select chaperones resulting in drastically accelerated client handoff. Perhaps DNA strand displacement occurs through an analogous mechanism, but the equivalent of strand displacement has not been discussed or defined for chaperone-client interactions, hence a new term.

Reviewer #2 (Remarks to the Author):

I'm grateful for the opportunity to review this interesting paper which casts light on TA-Sgt2 interactions through an

elegant, extensive and cohesive experimental program. This work will certainly be a valuable contribution to the current understanding of proteostasis. I include a few points for clarification below:

Towards the end of page 3 the paper states: 'Biochemical, structural, and yeast genetics studies showed that the major cytosolic Hsp70 (Ssa1 in yeast), with the help of Hsp40 (Ydj1 or Sis1 in yeast), effectively captures newly synthesized GET substrates and is important for maintaining them in a soluble, targeting-competent conformation 17–19. Hsp70 further initiates a sequential series of energetically downhill TA transfers.'

This is not the only interpretation of events surrounding the initial capture of TAs. Several studies indicate that Get4/5 complex binds directly to ribosomes in yeast (Ribosome-bound Get4/5 facilitates the capture of tail-anchored proteins by Sgt2 in yeast <https://pubmed.ncbi.nlm.nih.gov/33542241/> & Cotranslational Intersection between the SRP and GET Targeting Pathways to the Endoplasmic Reticulum of *Saccharomyces cerevisiae* <https://www.ncbi.nlm.nih.gov/pmc/articles/PMC5007794/>) and in the equivalent mammalian system SGTA is reported to have direct interaction with nascent TAs in the ribosomal tunnel even when it bears mutations that negate its ability to bind to Hsp70/40 (SGTA associates with nascent membrane protein precursors <https://pubmed.ncbi.nlm.nih.gov/32216016/>).

While ribosome association of Sgt2/SgtA/Get4/5 has been reported, the contribution of this association to TA targeting is unclear. None of these studies showed that disruption of the ribosome association reduces TA capture by Sgt2/SgtA/Get3 and/or compromises TA insertion. In contrast, the functional relevance of substrate capture and transfer through Hsp70 is more strongly established. Point mutations that disrupt the association of Hsp70 with Sgt2 strongly reduced TA capture by Sgt2 *in vitro* and *in vivo*, and rapid depletion of Hsp70 strongly disrupted TA insertion *in vivo*. We added references to these alternative models of TA loading in the Introduction, but do not want to leave the impression that these different models are equivalent in supporting evidence.

The results presented in the current manuscript also point towards flexibility in the order of things e.g. from page 10: '...the TA substrate alone increased the population of closed Sgt2 from 18% to 38% (Fig. 4A and 4D, dark green; Table S1). This indicates that the TA substrate can by itself drive substantial Sgt2 closing, and reciprocally, closed Sgt2 binds TA with higher affinity than open Sgt2.' The ambiguity (and likely flexibility) of this situation should be explained with references included to support all available perspectives, especially considering this is rightly done in the case of other controversial aspects e.g. the substrate interaction kinetics.

This is based on thermodynamic coupling of the substrate binding and conformational equilibria, shown in the following:

Here, E_{open} and E_{closed} refer to Sgt2 in the open and closed conformation, respectively, K_1 and K_4 are the TA binding affinities for open vs closed Sgt2, and K_3 and K_2 are the equilibria of the open-to-closed conformational change in free Sgt2 and TA-bound Sgt2, respectively. Because thermodynamics is pathway independent, we have:

$$K_1 \times K_2 = K_3 \times K_4$$

Rearranging this, one gets:

$$K_4 / K_1 = K_2 / K_3$$

This equation says that stronger substrate binding to closed Sgt2 than open Sgt2 is equivalent to more favorable closing of Sgt2 upon substrate binding, and vice versa. This equivalence is not due to ambiguity in mechanistic understanding but rather, to the fundamental property of thermodynamic coupling.

Further, in the middle of page 4: '...raising questions about whether both SBDs in the Sgt2 dimer are required for TA binding. However, previous SAXS (small-angle X-ray scattering) and NMR analyses suggested that the Sgt2 dimer adopts a wide-open conformation in which the two SBDs are predicted to be 130 Å apart...' This is true but neglects to mention a highly pertinent paper in which full-length SGTA has been shown to display open and closed conformations relating to its substrate interactions (Structural complexity of the co-chaperone SGTA: a conserved C-terminal region is implicated in dimerization and substrate quality control <https://pubmed.ncbi.nlm.nih.gov/29996828/>). There are two other important recent papers on the C-terminal

domain omitted from this account which should be discussed either in the introduction or discussion to give a complete picture about what is currently known of this elusive domain (Structural insights into metazoan pretargeting GET complexes <https://pubmed.ncbi.nlm.nih.gov/34887561/> Insights into the structure and function of the C-terminus of SGTs (small glutamine-rich TPR-containing proteins): A study of the Aedes aegypti homolog <https://pubmed.ncbi.nlm.nih.gov/34082040/>).

Thanks for the reminder and literature pointer. We have added these references.

The methods need more details. On page 18 the protein production for a number of different proteins is listed as 'described previously' with a large selection of papers cited and no indication of which reference refers to which protein. If someone were to wish to reproduce specific proteins from this study they would be sent down a rabbit hole of research and even then could not be sure if they were finding the correct protocol. To be able to compare this study accurately with others in the literature it is very important to know how these (some of them potentially challenging) proteins were prepared – particularly things like whether reducing conditions and/or detergents were used at any stage. Regarding the construct Sgt2 Δ N, I can't tell where the construct starts in the sequence and I also can't see evidence (apologies if I've missed this) that the protein remains stable throughout the experiments. Maybe it is fine but if it aggregated or degraded that would also account for the TA binding result. It seems plausible that, as suggested, the 2 C-terminal domains would need to be secured by the obligate N-terminal dimerization domains for efficient grasping of both sides of a TA but efficient binding between the isolated C-terminal domain and TAs is well documented (most recently in Molecular basis of tail-anchored integral membrane protein recognition by the cochaperone Sgt2 <https://pubmed.ncbi.nlm.nih.gov/33610544/> which is referenced elsewhere in the manuscript) – maybe the attached TPRs flopping around impede the TA binding of the Sgt2 Δ N construct but this should be discussed.

We have edited the Methods section with specified references for each protein. Most proteins were purified through standard His6 affinity columns, and their purification was described in detail before (Cho et al, EMBO 2018 and Rao et al, eLife 2016, respectively). Since Sgt2 is prone to degradation during mechanical cell disruption, we used a mild detergent lysis method (as described in Rao et al, eLife 2016), and this detail was added to the Methods.

Lin et al studied TA interaction with monomeric Sgt2 using Sgt2-TPR-C, which is the same as the Sgt2 Δ N construct we used here (we have added the sequence information in the method section). Compared to the work in Lin et al, we examined TA-Sgt2 interaction under more stringent conditions. Lin et al used Sgt2 and TA variants that are either co-overexpressed in E. coli or at high sufficiently concentrations to appear as overloaded bands in Coomassie-blue stained gels (they did not give information on protein concentrations, but judging from the gels, protein concentrations are in the high micromolar to tens of micromolar range). In vivo, Sgt2 concentration is ~500 nM, and we used near physiological concentrations of Sgt2 in most of our experiments. In addition, pulling on TA instead of Sgt2 (Figure 2) also sensitizes the system to differences in TA interaction stability.

In the initial chase experiments on page 6, the authors find that unlabeled Sgt2 can displace TA binding at a similar rate to CaM which makes me wonder whether the equivalent competition was tried in the ALEX experiments (in which Sgt2 was differentially FRET labeled on either side of the dimer - page 9) to see whether adding unlabeled Sgt2 would similarly displace the TA and veer the labeled Sgt2 back towards favoring the open conformation.

We did the suggested experiment (mixing donor-labeled Sgt2 with acceptor labeled Sgt2) and did not observe monomer exchange between Sgt2 dimers, as would be indicated by colocalization of both dyes in the same particle. This indicates that Sgt2 dimers are stable, and TAs are exchanged between Sgt2 dimers but not with Sgt2 monomers.

The crosslinking experiment in Figure S7 also indicates the absence of monomer exchange between Sgt2 dimers. If exchange occurred, we should have observed crosslink between unlabeled and BDP-labeled Sgt2.

I must state that I am not experienced with many of the techniques used in this paper but the authors explain them admirably well and the interpretations of results seem justified. I would say that that sometimes more caveated language should be used as the effects, especially with regard to the different TA variants that veer towards hydrophobicity are arguable somewhat subtle.

Please see our responses to comments 2 and 8 by reviewer #1. To summarize, the smFRET data that we interpreted on had large number of replicates and data points to give confidence. We also repeated the measurements with the 6AG, which upon closer inspection of the raw data may have problems with sample aggregation. The new data now has a total of 10 replicates and show that with 6AG, Sgt2 closing equilibrium was reduced to the level of the equivalent sample without TA.

I'm not sure about the word privilege – it occurs 15 times in the manuscript – instinctively it doesn't come across as quite the right word – but maybe I'm just unfamiliar with this usage and I'm very much willing to concede to the opinions of other reviewers on this. I think I get what the authors mean – that the system prioritizes more hydrophobic TAs for directional targeting to the GET pathway but privilege suggests something like positive discrimination to me whereas I envisage Sgt2 to be sorting substrates and sending them to different fates, rather than preferentially picking the ones that exceed a hydrophobicity threshold – maybe this is more true of SGTA though which associates with more complex machinery than the yeast system.

We indeed use 'privilege' to denote that the transfers within this chaperone triad is a positive selection, because both transfers are >100-faster than spontaneous dissociation and moreover, Get3 selectively captures TAs from closed Sgt2 dimer in which the TA is strongly shielded from other chaperones. Nevertheless, the word was repeated too many times in the manuscript, and we have removed some of them.

Finally a small formatting thing - for lists of sequences e.g. in Fig S8A it's much easier to look at if Courier font is used so that each letter takes up the same amount of space and the alignment is clearer.
Thanks a lot for the suggestion. We have changed it accordingly.

Reviewers' Comments:

Reviewer #1:

Remarks to the Author:

I thank the authors for addressing this reviewer's requests and for making the additional experiments. However, in my view, there are still several points unclear and a few things are even wrong. Therefore, I cannot support publication of the manuscript in its current form.

Most critical are the following points:

1) I still do not understand, why the authors do not want to show the raw donor and acceptor data for Fig. 1B,C,F, 6C, 7C or alternatively provide the data in a supporting file, then I could have a quick look and maybe try to fit the data myself. I think that it should be general practice to show or provide raw data in the supplement or as an additional file. Here, this is very important, as the absolute fluorescence of the donor at $t=0$ (which is never shown) has to be about the same for each of the experiments. In addition, it is crucial to see the amount of change relative to the baseline to judge the quality of the signal and to spot potential artifacts.

This would then probably answer all my still open questions, because I still do not understand what is "normalized fluorescence change" exactly. How is the normalization done (linear, linear + offset, ...) – please provide the formula.

Then, I do not agree with your definition of the FRET efficiency E . In the formula from Lakowicz you cite, F is the fluorescence lifetime of the dye (once in presence of the acceptor and once in absence of the acceptor), but the authors measure the fluorescence intensity. Therefore they use the wrong formula, the FRET efficiency for intensities is defined as $E=IA/(IA+\gamma>ID)$ therefore the acceptor intensity IA is crucial.

Finally, all the fit values and not only the rate constants should be given, e.g. F_e and ΔF when equation 1 is used. This would help to clarify all quantities used, because some are still not clear. I thought that the normalized fluorescence change is normalized to 1 at the end of the reaction and that this is when the reaction reaches equilibrium (i.e. F_e). But if F_e is one, then all values in equation 1 (and also equation 2) will be larger than 1 (as the exponential function is always positive), which clearly cannot fit the data.

2) The FRET efficiency and stoichiometry from microsecond ALEX were not corrected for leakage, direct excitation and gamma factor. The authors do correctly state, that this lack of correction does not affect general trends, but the absolute values are often significantly affected. This is particularly relevant for the selection of states by absolute values of stoichiometry, which the authors have done by using $0.2 < S < 0.8$. When I see this uncorrected data, I would first expect state 3 to be donor only (although the authors have applied a filter for donor only, in my view the filter through absolute stoichiometry is best and usually done). This would completely change the analysis and results, because instead of three states there would be only two states (and donor only molecules, which do not report on a state of Sgt2). If state 3 really should be a conformational state of Sgt2, how can the stoichiometry be explained? The location of the state would require s.th. like 2 or 3 times more donor molecules than acceptor molecules in one molecule, which would hint towards overlabelling with donor dyes, which then again would lead to protein species, which in my view should not be considered for kinetic analysis of Sgt2's conformational changes.

3) I had a quick look at the comments from reviewer #2 and although I do usually not comment on replies to other reviewers, this time I have to, because there is a fundamental error: The authors claim that $K1 \times K2 = K3 \times K4$ because thermodynamics is pathway independent. This is wrong. Thermodynamics does not say anything about the rates (only about ratios of forward and backward rates), because thermodynamics it is only dependent on the free energy difference and not the barrier height (which defines the rates). Just think of a simple example, where on one pathway the barriers are very low and on the other pathway the barriers are high, while the equilibrium energies are the same. Then thermodynamics is the same on both paths, but you will clearly see that this equation ($K1 \times K2 = K3 \times K4$) does not hold.

Reviewer #2:

Remarks to the Author:

My comments have been addressed adequately. I'm not as qualified to judge the responses to the comments of the other reviewer but they seem okay and if the other reviewer has no further issues then I'd be happy to see this published.

Reviewer #1 (Remarks to the Author):

I thank the authors for addressing this reviewer's requests and for making the additional experiments. However, in my view, there are still several points unclear and a few things are even wrong. Therefore, I cannot support publication of the manuscript in its current form. Most critical are the following points:

1) I still do not understand, why the authors do not want to show the raw donor and acceptor data for Fig. 1B,C,F, 6C, 7C or alternatively provide the data in a supporting file, then I could have a quick look and maybe try to fit the data myself. I think that it should be general practice to show or provide raw data in the supplement or as an additional file. Here, this is very important, as the absolute fluorescence of the donor at $t=0$ (which is never shown) has to be about the same for each of the experiments. In addition, it is crucial to see the amount of change relative to the baseline to judge the quality of the signal and to spot potential artifacts. This would then probably answer all my still open questions, because I still do not understand what is “normalized fluorescence change” exactly. How is the normalization done (linear, linear + offset, ...) – please provide the formula.

We have placed all the raw donor fluorescence time traces for Figs 1B, C, F, 6C and 7C in Appendix 0, which is uploaded as a supplemental file. Note that we always measured the initial donor fluorescence ($F_{t=0}$) in each experiment before mixing with chaperone, and the mixing deadtime was recorded and added to the total time of data collection.

Normalized fluorescence change = $(F-F_0)/(F_e-F_0)$, where F is observed fluorescence intensity, F_0 and F_e are fluorescence intensity at $t = 0$ and when the reaction plateaued, respectively. We have defined this in Eq 1 and relabeled the concerned figures with this term.

We in general do not follow time traces of acceptor fluorescence change for reasons explained in the previous response: acceptor-labeled chaperone is in excess over donor-labeled substrate, so the fraction of acceptor that is bound to substrate and can change fluorescence is small. This can be seen in Figure Response-1. In this experiment, we recorded the fluorescence spectra of the sample before and after the chase reaction reached equilibrium (B). A decrease in acceptor fluorescence can be observed as well as a rise in donor fluorescence (which we monitored in A). However, the % acceptor fluorescence change is small and hence not used.

Figure response-1

A Donor fluorescence intensity before and after addition of unlabeled Sgt2

B Fluorescence spectra before and after addition of unlabeled Sgt2

We also note that the absolute donor fluorescence intensity at $t = 0$ is not expected to be the same across all experiments. This is because fluorescence intensity is dependent on instrument setting, concentration of labeled sample, labeling efficiency, etc. There are many fluorescence measurements in this work, which were collected with independently prepared samples, measured on different days (some years apart from each other), on two different fluorimeters, and with lamps at different stages of lifetime and hence give off different amounts of excitation light. Some variations are expected, though fluorescence values are mostly in the same range. Importantly, the kinetics of the reactions is independent of sample concentration or fluorescence signal.

Then, I do not agree with your definition of the FRET efficiency E . In the formula from Lakowicz you cite, F is the fluorescence lifetime of the dye (once in presence of the acceptor and once in absence of the acceptor), but the authors measure the fluorescence intensity. Therefore they use the wrong formula, the FRET efficiency for intensities is defined as $E = I_A / (I_A + \gamma \cdot I_D)$ therefore the acceptor intensity I_A is crucial.

Below is p370 of the Lakowicz book, which states that FRET can be measured using either donor intensity change (Eq 13.14), or lifetime change (Eq 13.13). Either way, *donor fluorescence* is monitored in bulk FRET measurements. Calculating FRET efficiency based on acceptor fluorescence intensity is a practice in single molecule FRET measurements due to complications of donor photobleaching, and we calculated E^* (effective FRET efficiency) this way in our μ s-ALEX measurements (see eq 6).

$$E = \frac{R_0^6}{R_0^6 + r^6} \quad [13.12]$$

This equation shows that the transfer efficiency is strongly dependent on distance when the D–A distance is near R_0 (Figure 13.2). The efficiency quickly increases to 1.0 as the D–A distance decreases below R_0 . For instance, if $r = 0.1R_0$, one can readily calculate that the transfer efficiency is 0.999999, so that the donor emission would not be observable. Conversely, the transfer efficiency quickly decreases to zero if r is greater than R_0 . Because E depends so strongly on distance, measurements of the distance (r) are only reliable when r is within a factor of 2 of R_0 (see Problem 13.7). If r is twice the Förster distance ($r = 2R_0$), then the transfer efficiency is 1.56%.

The transfer efficiency is typically measured using the relative fluorescence intensity of the donor, in the absence (F_D) and presence (F_{DA}) of acceptor. The transfer efficiency can also be calculated from the lifetimes under these respective conditions (τ_D and τ_{DA}):

$$E = 1 - \frac{\tau_{DA}}{\tau_D} \quad [13.13]$$

$$E = 1 - \frac{F_{DA}}{F_D} \quad [13.14]$$

It is important to remember the assumptions involved in the derivation of these equations. Equations [13.13] and [13.14] are only applicable to donor–acceptor pairs which are separated by a fixed distance. This situation is frequently encountered for labeled proteins. However, a single fixed donor–acceptor distance is not found for a

mixture of donor and acceptors complex expressions are rate over the acceptor pairs.⁵

The use of lifetime measurements is complicated by spectral overlap and energy transfer. In the presence of acceptor, the lifetime of the donor is shortened (τ_{DA}) because of energy transfer. This is rare in biomolecular systems because of the exponential dependence of the transfer rate on distance. The transfer rate is only significant when the donor–acceptor distance is within a factor of 2 of R_0 . The transfer rate can be expressed as

Assuming that the donor and acceptor are separated by a distance r , one sees that the transfer rate is proportional to R_0^6/r^6 , which in turn depends on the refractive index of the medium. The refractive index of the medium is a function of composition or refractive index ($n = 1.33$) or scattering cross-section. The quantum yield of the donor is a function of the refractive index and the quantum yield of the donor with standard deviation. The transfer rate is proportional to the sixth root in the refractive index. The transfer rate is not have a large effect on the transfer rate. The transfer rate is evaluated for each donor–acceptor pair. The transfer rate is evaluated for each emission spectrum of the acceptor. The transfer rate is evaluated for each acceptor with large R_0 values. In the ex-

Finally, all the fit values and not only the rate constants should be given, e.g. F_e and $\Delta\delta_F$ when equation 1 is used. This would help to clarify all quantities used, because some are still not clear. I thought that the normalized fluorescence change is normalized to 1 at the end of the reaction and that this is when the reaction reaches equilibrium (i.e. F_e). But if F_e is one, then all values in equation 1 (and also equation 2) will be larger than 1 (as the exponential function is always positive), which clearly cannot fit the data.

There were some typos in Eq 1 and Eq 2 (a mix-up of typing equations for exponential decay vs exponential rise), which we now fixed. We thank the reviewer for catching this. The correct equations were used during data analysis and fitting (otherwise nothing would have fit). We showed the fit values for all the parameters in the new Appendix 0.

2) The FRET efficiency and stoichiometry from microsecond ALEX were not corrected for leakage, direct excitation and gamma factor. The authors do correctly state, that this lack of correction does not affect general trends, but the absolute values are often significantly affected. This is particularly relevant for the selection of states by absolute values of stoichiometry, which the authors have done by using $0.2 < S < 0.8$. When I see this uncorrected data, I would first expect state 3 to be donor only (although the authors have applied a filter for donor only, in my view the filter through absolute stoichiometry is best and usually done). This would completely change the analysis and results, because instead of three states there would be only two states (and donor only molecules, which do not report on a state of S_{gt2}). If state 3 really should be a conformational state of S_{gt2}, how can the stoichiometry be explained? The location of the state would require s.th. like 2 or 3 times more donor molecules than acceptor molecules in one molecule, which would hint towards overlabelling with donor dyes, which then again would lead to protein species, which in my view should not be considered for kinetic analysis of S_{gt2}'s conformational changes.

Please note that the E-S plots generated by mpH²MM analyses are for states' dwells, not bursts. They are denoted as S_{dwell} and E_{dwell} . The data were first analyzed using dual-channel burst search that identifies bursts with the correct stoichiometry (burst-selection criteria described in Methods). mpH²MM then identifies and characterizes states and their transitions within individual selected bursts (E. Nir et.al., *J. Phys. Chem. B*, **110**, 22103; A. Ingargiola et.al., *PLoS ONE*, **11**(8): e0160716). The identified states within a burst are termed dwells. We made a new Figure S5 that shows the data analysis work-flow of the μ s-ALEX experiments to help readers better understand the process.

As the reviewer said and as we indicated in the manuscript, State 3 (the population with low E_{dwell} & high S_{dwell}) is not related to the conformational state of S_{gt2}, but rather is caused by dye photophysics. While we acknowledge the possibility that state 3 could be contaminated by donor-only species, donor-only bursts are mostly removed after the dual channel burst search. We therefore expect that state 3 was primarily due to dye-photophysics, such as acceptor blinking events on the sub-millisecond scale. Most importantly, our model contains only two conformational states, open (state 1) and closed (state 2), which were used exclusively to calculate parameters of the conformational changes of S_{gt2} after mpH²MM analysis. State 3, once identified, was excluded from the calculation.

3) I had a quick look at the comments from reviewer #2 and although I do usually not comment on replies to other reviewers, this time I have to, because there is a fundamental error: The authors claim that $K_1 \times K_2 = K_3 \times K_4$ because thermodynamics is pathway independent. This is wrong. Thermodynamics does not say anything about the rates (only about ratios of

forward and backward rates), because thermodynamics it is only dependent on the free energy difference and not the barrier height (which defines the rates). Just think of a simple example, where on one pathway the barriers are very low and on the other pathway the barriers are high, while the equilibrium energies are the same. Then thermodynamics is the same on both paths, but you will clearly see that this equation ($K_1 \times K_2 = K_3 \times K_4$) does not hold.

It appears that the reviewer and I are saying the same things: thermodynamics is pathway-independent. $K_1 \times K_2 = K_3 \times K_4$ is simply a mathematical statement of this principle. Both in the previous response letter and in the scheme to illustrate the situation (copied below), we used upper-case 'K's, which denote equilibrium constants. Rate constants would have been denoted by lowercase 'k's. Our point, in the previous letter, was simply that more favorable closing observed for Sgt2-TA compared to Sgt2 implies that TA binds closed Sgt2 more strongly than open Sgt2, and vice versa, because these two equilibria are thermodynamically linked. These are strictly thermodynamic descriptions. Nowhere in the letter or the manuscript did we try to distinguish whether Sgt2 closes first followed by TA binding, or if TA binds open Sgt2 first followed by Sgt2 closing.

Reviewer #2 (Remarks to the Author):

My comments have been addressed adequately. I'm not as qualified to judge the responses to the comments of the other reviewer but they seem okay and if the other reviewer has no further issues then I'd be happy to see this published.

Reviewers' Comments:

Reviewer #1:

Remarks to the Author:

I thank the authors for addressing this reviewer's final points, in particular thanks for:

- 1) Providing the raw data for several figures, which actually looks very good.
- 2) For explaining why the absolute donor fluorescence intensity at $t = 0$ is not expected to be the same across all experiments and explaining the calculation of the FRET efficiency.
- 3) For providing the correct equations for fitting.
- 4) For explaining the μ s-ALEX experiments in more detail.
- 5) For clarifying that state3, once identified, was excluded for rate constant calculations.
- 6) For clarifying that you did not mean to claim that Sgt2 closes first (after Hsp70 binding) followed by TA binding to the closed state. I likely misinterpreted the little arrow from TA to closed Sgt2 and the word 'handover'.

Overall, I now support publication of the manuscript.